# Using paired catchments to quantify the human influence on hydrological droughts

Anne F. Van Loon[1,*], Sally Rangecroft[1,*], Gemma Coxon[2], Jose Agustin Breña-Naranjo[3], Floris Van Ogtrop[4] & Henny A. J. Van Lanen[5]

[1] School of Geography, Earth and Environmental Sciences, University of Birmingham, Birmingham, UK

[2] School of Geographical Sciences, University of Bristol, UK

[3] Institute of Engineering, Universidad Nacional Autónoma de México, Mexico

[4] School of Life and Environmental Sciences, University of Sydney, Sydney, Australia

[5] Hydrology and Quantitative Water Management Group, Wageningen University, The Netherlands

* These authors contributed equally to this research.

*Correspondence to*: Anne Van Loon (a.f.vanloon@bham.ac.uk)

**Abstract.** Quantifying the influence of human activities, such as reservoir building, water abstraction, and land use change, on hydrology is crucial for sustainable future water management, especially during drought. Model-based methods are very time-consuming to set up and require a good understanding of human processes and time series of water abstraction, land use change and water infrastructure and management, which often are not available. Therefore, observation-based methods are being developed that give an indication of the direction and magnitude of the human influence on hydrological drought based on limited data. We suggest adding to those methods a 'paired-catchment' approach, based on the classic hydrology approach that was developed in the 1920s for assessing the impact of land cover treatment on water quantity and quality. When applying the paired-catchment approach to long-term pre-existing human influences trying to detect an influence on extreme events such as droughts, a good catchment selection is crucial. The disturbed catchment needs to be paired with a catchment that is similar in all aspects except for the human activity under study, in that way isolating the effect of that specific activity. In this paper, we present a framework for selecting suitable paired catchments for the study of the human influence on hydrological drought. Essential elements in this framework are the availability of qualitative information on the human activity under study (type, timing and magnitude), and similarity of climate, geology, and other human influences between the catchments. We show the application of the framework on two contrasting case studies, one impacted by groundwater abstraction and one with a water transfer from another region. Applying the paired-catchment approach showed how the groundwater abstraction aggravated streamflow drought by more than 200% for some metrics (total drought duration and total drought deficit) and the water transfer alleviated droughts with 25 to 80%, dependent on the metric. Benefits of the paired-catchment approach are that climate variability between pre- and post-disturbance periods does not

have to be considered as the same time periods are used for analysis, and that it avoids assumptions considered when partly or fully relying on simulation modelling. Limitations of the approach are that finding a suitable catchment pair can be very challenging, often no pre-disturbance records are available to establish the natural difference between the catchments, and long time series of hydrological data are needed to robustly detect the effect of the human activities on hydrological drought.

We suggest that the approach can be used for a first estimate of the human influence on hydrological drought, to steer campaigns to collect more data, and to complement and improve other existing methods (e.g. model-based or large-sample approaches).

# 1 Introduction

## 1.1 Background

In our human-modified era, the Anthropocene, human activities have direct and indirect effects on the hydrological system (UNESCO, 2009; 2012; Montanari et al., 2013; Destouni et al., 2013; McMillan et al., 2016). It is vital to understand how our activities are affecting the hydrological system to help us improve the management of water resources, especially during drought. Only limited studies exist that focus on the quantification of how humans influence hydrological droughts (Querner et al., 1997; Querner & Van Lanen, 2001; Van Lanen et al., 2004b). This has inspired recent calls for new tools and

approaches to study the human influence on droughts (Van Loon et al., 2016a; 2016b). Here we define drought as a deficit in available water from 'normal' conditions (Wilhite & Glantz, 1985; Tallaksen & Van Lanen, 2004) and we focus on hydrological drought, which is a deficit in streamflow.

Recent studies quantifying the human influence on hydrological droughts use hydrological models because of their ability to isolate processes by generating scenarios with and without human activities for comparison (e.g. Wanders & Wada, 2015;

Van Loon & Van Lanen, 2015; Veldkamp et al., 2015; Wada et al., 2017). Also in other fields, such as hydroecology, modelling is used to quantify hydrologic alteration, i.e. the deviation of flows between actual and baseline conditions (e.g. Poff et al., 2010; Mathews and Richter, 2007). Often this 'scenario modelling' is done with large-scale models (e.g. Veldkamp et al., 2015; Wanders & Wada, 2015; Wada et al., 2017), but these models have a coarse resolution and are not extensively calibrated or validated locally, and will therefore have large uncertainty on the local scale. Furthermore, whilst

the scientific community is working to add known human activities and decisions into hydrological models (Wada et al., 2017), the current generation of hydrological models does not include all anthropogenic processes yet (Srinivasan et al., 2017) and models often underestimate the effect of these processes (Jaramillo and Destouni, 2015). Finally, scenario modelling requires high-resolution data on human activities influencing the hydrology of a catchment (e.g. land use, abstraction), which often are not available. Therefore, we need relatively simple methods that can estimate the influence of

human activities on drought from observational data. Several methods have been developed in recent years:

(1) the 'observation-modelling' approach (e.g. Van Loon & Van Lanen, 2013) compares human-influenced observed droughts with naturalised simulated droughts. Downsides of this method are that the simulated hydrological data have uncertainties and a pre-disturbed period is needed for calibration to reduce those;

(2) the 'upstream-downstream' approach (Rangecroft et al., 2019) compares hydrological droughts downstream of a disturbance with those upstream, which are assumed to be unaffected. This method uses observation data only and uncertainty comes from the possible non-linear relationship between the upstream and downstream gauging stations;

(3) the 'pre-post-disturbance' approach (e.g. Liu et al., 2016) compares hydrological droughts before and after a disturbance. The comparison of two different time periods makes it harder to separate the human influence from climatic variability (e.g. decadal and multi-decadal) and non-stationarity due to climate change (Peñas et al., 2016);

(4) and finally, the 'large-scale screening' approach uses comparative hydrology (Wagener et al., 2010) or large-sample hydrology (Gupta et al., 2014) to disentangle different drivers of hydrological processes. For example, Destouni et al. (2013) relate changes in evaporation to hydro-climatic factors and land & water use in Sweden, and Tijdeman et al. (2018) use deviations in the relationship of meteorological to hydrological drought between human-influenced catchments and a range of benchmark catchments in the UK. Downside of this method is the need for a large number of catchments with long time series of hydrological data and information about the type and degree of human influence for all catchments.

In this paper, we suggest to add an observation-based approach that requires very little data and uses the same time period for the analysis, thereby avoiding effects of climatic variability and non-stationarity. The 'paired-catchment' approach compares a human-influenced catchment with a benchmark catchment where the human activity of interest is not present. Paired-catchment analysis is a classical method in hydrology that compares the flow regime of two catchments which have similar physical characteristics with the aim to identify the impact of a disturbance on the flow regime (Hewlett, 1971; Bosch & Hewlett, 1982). The approach has been applied around the world to evaluate and quantify effects of land use change and treatment (e.g. afforestation, deforestation) on hydrology (e.g. water yields and water quality) (Brooks et al., 2003; Brown et al., 2005; Folton et al., 2015). There are some paired-catchment studies focusing on low flows, but no applications investigating human influence on hydrological drought. We argue that this paired-catchment approach could help quantify the human influence on hydrological drought and complement existing approaches.

## 1.2 Paired-catchment analysis

The paired-catchment approach is a classic method for detecting the effects of disturbance on catchment hydrology (Bates, 1921; Zégre et al., 2010). It originates from experimental research on the effects of intentional land cover treatment (e.g. afforestation, deforestation) on water quantity and quality. The basic concept of the method is to compare the flow regime of two nearby catchments with similar physical characteristics, one as a control ('benchmark') and the other as a disturbed catchment (also known as the 'treatment catchment' in some of the literature). Comparing the same time periods allows for climatic variability to be accounted for in the analysis (Brown et al., 2005). Climate, soils and geology should be similar between the two catchments (Best et al., 2003; Brown et al., 2005; Folton et al., 2015), with the main difference the

treatment in the disturbed catchment. Traditionally, this treatment often consists of deforestation or afforestation. The identified differences in hydrology between the disturbed and control catchments can then be attributed to the treatment (Brown et al., 2005). In most paired-catchment studies, catchments are typically adjacent, although this is not always possible, but they tend to be in close proximity to help with the similarity of catchment characteristics and climate.

Relatively short time periods are used for the analysis of the effects of the treatment, typically a few months to years. For classic paired-catchment treatment studies, pre-disturbance periods are sometimes used for both catchments to ensure streamflow similarity, or to establish the pre-disturbance difference in hydrology between the catchments. This set up is often called "Before-After Control–Impact" (BACI), which is different from the simpler "Before–After" (BA; approach 3 mentioned above) and "Control–Impact" (CI) set ups (Peñas et al., 2016).

The paired-catchment approach has been used for many decades (Swank et al., 2001; Brooks et al., 2003; Brown et al., 2005; Putro et al., 2016). Starting as early as the 1920s (Bates, 1921), the use of paired catchments to study the impacts of forest treatments and management activities on water yields accelerated since the 1960s (e.g. Hewlett & Hibbert, 1961; Harris, 1977; Hornbeck et al., 1993; Robinson & Rycroft, 1999). There is some controversy around the trustworthiness of paired-catchment results. Some studies have shown that the paired-catchment approach has a better performance than standard

calibrated rainfall-runoff models (e.g. Andréassian et al., 2012), but some critique the way paired-catchment observations are analysed (e.g. Alila et al., 2009) or the way results are interpreted (e.g. Calder & Aylward, 2006). These controversies influence the societal debate around the effectiveness of catchment management, especially on the effects of forests on flooding (Andréassian, 2004; Ellison et al., 2012).

Overall, most published studies have looked at land use change impacts on annual flow and flood peaks and assessing the

magnitude of water yield change resulting from changes in vegetation (see the review paper of paired-catchment analysis by Brown et al., 2005 and other studies by Cornish, 1993; Bari et al., 1996; Best et al., 2003; Folton et al., 2015). In the 1990s, some studies started looking at low flows within paired-catchment analysis (e.g. Keppeler & Ziemer, 1990; Scott & Smith, 1997). Low flows are defined as the "minimum flow in a river during the dry periods of the year", which makes them a seasonal phenomenon and an integral component of the flow regime of a river (Smakhtin, 2001). It has for example been

found that clearcut harvesting can lead to an increase in low flows (Keppeler & Ziemer, 1990), while conversion from grasslands, shrublands and croplands to forests can cause a decrease of low flows (Scott & Smith, 1997; Farley et al., 2005). Paired-catchment studies focusing on low flows also suggested that in watersheds located in dry regions streams were likely to completely dry up following afforestation, and that the streamflow regime in those watersheds would change from perennial to intermittent (Farley et al., 2005; Jackson et al., 2009). Droughts differ from low flows because they represent an

anomaly from the normal seasonal cycle and can therefore also occur in the high flow season. There currently is no research using the paired-catchment approach to assess changes in hydrological droughts due to land use change and other human activities.

Because droughts are extreme events that occur irregularly, long time series are needed to detect any effect of human influences on hydrological drought using a paired-catchment approach. Additionally, many human activities cannot be

applied as intentional treatment in a small catchment just for research purposes. This is the case for example with reservoirs, groundwater abstraction and other large-scale water supply and management activities. If we want to study the effects of these on hydrological drought, we need to work with existing observed data. Often no hydrological monitoring was done before the start of the human activity, which means that pre-disturbance data is not available to assess catchment similarity

and a full BACI set up is not possible. Despite this, the paired-catchment approach can still be used with a CI set up, if the selection of the catchment pair is done carefully and qualitative data on the human activities is available. The CI set up has been found to give satisfactory results for a range of hydrological variables compared to the baseline BACI set up that uses paired catchments that are calibrated on a pre-disturbance period (Peñas et al., 2016). A long-term paired-catchment study using the CI set up has recently been done to assess the influence of urbanisation on flooding in the UK (Prosdocimi et al.,

2015) and we think that there is potential also to use it in drought studies.

### 1.3 Aim

In this paper, we explore the use of the paired-catchment approach to assess the human influence on hydrological droughts.

We present two case studies, from the UK and Australia, as examples of how this approach can help to generate knowledge on the human influence on hydrological droughts from observation data at the catchment scale. The two contrasting case studies used here have water added (via water transfer) or removed (via groundwater abstraction) in the human-influenced catchment. By showing the application of the paired-catchment approach to drought and discussing its limitations, we demonstrate how this approach can be used as tool for improved analysis and understanding of the human influence on

hydrological drought in the Anthropocene.

### 2. Application of the paired-catchment approach to quantify the human influence on hydrological drought

For applying the paired-catchment approach to quantify the human influence on hydrological drought, two catchments need to be selected, of which one is a human-influenced catchment (e.g. with water abstraction, urbanisation, reservoirs) and the

other is as similar as possible except that it does not have the human influence we aim to quantify. Importantly, the benchmark catchment does not need to be completely natural to be used in a paired-catchment setting as long as the human influence of interest can be isolated and quantified. The analysis focuses on the effect on hydrological drought metrics (e.g. drought duration and deficit volume), rather than the more typical focus of paired-catchment studies on water yield or low flows. Therefore, long time series are needed to extract these drought metrics. It also avoids the problematic one-to-one

pairing of events based on driving meteorology (chronological pairing), because results would be too dependent on event characteristics and antecedent conditions (Alila et al., 2009). Instead we focus on the whole time series, taking differences in both frequency and magnitude into account (frequency pairing), as was advocated by Alila et al. (2009). In this section, we

outline the important elements for the approach, such as the catchment selection process, the data requirements, the drought analysis, and the calculation for quantifying the human influence.

## 2.1 Catchment selection

The starting point for a paired-catchment analysis is a human activity that is expected to influence the hydrology of an area.

Qualitative information about this human influence (such as type and timing) and long time series of hydrological data need to be available (Fig. 1). One of the most critical requirements for a successful paired-catchment comparison is that the catchment characteristics are as similar as possible, except for the human disturbance that is being analysed. Relevant catchment characteristics could include precipitation, temperature or potential evapotranspiration, land use, catchment area, elevation, geology, and soils. How similar these characteristics should be for a paired-catchment analysis will vary

depending on the hydrological extreme of interest (e.g. floods, droughts) and the dominant characteristics that control hydrological response in that region (e.g. potential evapotranspiration may be more important in drier, hotter climates; Oudin et al., 2010). There is little literature on the acceptable differences between the paired catchments, which makes it difficult to automatise the process. Paired catchments are therefore often chosen manually using expert judgement.

In Fig. 1, we provide some guidance on which characteristics are important for this application of the paired-catchment

approach to drought, and how these could be assessed. For drought studies, precipitation, potential evapotranspiration, soils, geology and land use are considered to be the most important characteristics (Van Lanen et al., 2004a; 2013; Stoelzle et al., 2014) to ensure similar meteorological conditions and hydrological response in both catchments. It is often quite difficult to find paired catchments with exactly the same characteristics so it is necessary to put some limits on what is deemed acceptable for pairing. In this study, catchments were considered to be suitable for pairing if differences in average annual

precipitation and potential evapotranspiration were within ±10% and the catchments had the same geological classification (Fig. 1). It is also important to check precipitation variability as well as annual values, as the annual averages could be masking very different distributions of precipitation which would then result in different meteorological droughts in the catchments. This check can be done by plotting the precipitation values for both catchments on the annual and monthly time scale to see their similarities and differences. Case studies where precipitation distribution between the catchments

significantly diverted from the 1:1-line or had linear regression slopes and $R^2$ values of less than 0.5 should be excluded from paired-catchment analysis. Soil type and geology are important to consider with regard to their responsiveness, i.e. flashy versus slow response to precipitation, implying that expert knowledge is needed to translate the qualitative soil and geology information. To enable isolating the effect of the human activity on drought, the catchment selection also requires a singular human influence that is different between the paired catchments. Similarity between the catchments with regard to other

human activities should be checked at least qualitatively.

For our application of the paired-catchment analysis there are some catchment characteristics that cannot be used as catchment selection criterion. For example, the Base Flow Index (BFI) calculated from discharge data cannot be used in catchment selection because the BFI of the human-influenced catchment is probably affected by the human activity. The

proximity of catchments a good starting point in the search for similar catchments. However, it may not be a useful catchment selection criterion because neighbouring catchments may have very different geology or precipitation totals (e.g. due to rain shadow effect) or because the human influence crosses catchment boundaries (e.g. groundwater abstraction, urbanisation).

## 2.2 Data requirements

Ideally, observation data of precipitation and discharge with a minimum of 30 years should be used, with limited missing data ($< 5\%$) so that hydrological extremes can be computed reliably (McKee et al., 1993; Rees et al., 2004). Information on catchment characteristics and the type of human activity in the human-influenced catchment are also required (Fig. 1). To check similarity on other human activities, information is needed on the presence of reservoirs, outflow of sewage treatment plants and other artificial drainage or water transfer, water abstraction for public water supply or irrigation.

Most commonly, the paired-catchment approach is done with annual data, but monthly data is also sometimes used (Bari et al., 1996; Brown et al., 2005). Droughts are generally long phenomena with timescales of weeks to years, with drought-generating processes and associated impacts going beyond a daily time step. Therefore drought analysis is normally conducted on the monthly time step (e.g. Hisdal et al., 2004; Fleig et al., 2006). Consequently, here we use monthly data for the paired-catchment analysis.

## 2.3 Drought analysis

Drought analysis can be done with several different methods (Van Loon, 2015). For paired-catchment analysis we suggest using the threshold level method (Yevjevich, 1967; Tallaksen & Van Lanen, 2004) to identify drought events and their metrics, because it allows to use the benchmark regime to compare the human-influenced flow against. The threshold level method is a commonly used drought analysis method that defines drought events when streamflow is below a specified threshold (Yevjevich, 1967; Hisdal & Tallaksen, 2000; Hisdal et al., 2004; Fleig et al., 2006). For quantification of the human influence, the threshold needs to be generated from the benchmark catchment time series (Fig. 2; Rangecroft et al., 2019), therefore excluding any influence of human activities on the threshold. This benchmark threshold is then applied to both catchments for the drought analysis (Fig. 2).

Different threshold levels can be used. In the case studies described here, we used the 80% non-exceedance threshold (Q80) from the flow duration curve. This means that 80% of the time discharge is above this threshold. The Q80 is a commonly used threshold to identify drought events (Hisdal & Tallaksen, 2000; Tallaksen and Van Lanen, 2004; Fleig et al., 2006; Heudorfer & Stahl, 2016). The threshold can be fixed or variable; we used the monthly variable threshold to incorporate seasonality into the threshold (Hisdal & Tallaksen, 2000; Hisdal et al., 2004; Fleig et al., 2006; Heudorfer & Stahl, 2016).

From the comparison between the flow and the threshold, several drought metrics can be calculated and when applying paired-catchment analysis it is important to consider more than one to avoid misinterpretation (Alila et al., 2009). Drought events are defined as consecutive periods of several months of below-threshold flow. We excluded drought events with a

duration of only 1 month from the analysis process so that all drought events are longer than the time step of the threshold. The first drought metric used for the paired-catchment comparison is 'occurrence of drought events'. We calculated the frequency, which is the total number of drought events identified in the time period. The second drought metric analysed is 'duration of drought events'. We used the total number of months in drought, the average duration of all events, and the duration of the maximum event. The third drought metric is 'deficit volume of drought events', which is defined as the cumulative difference between the flow and the threshold over the drought event. Again we used the total deficit over the entire time period, the average deficit of all events, and the deficit of the maximum event.

## 2.4 Estimation of the human impact on drought characteristics

Chronological pairing of flooding events is known to be difficult because storms do not always coincide in time, duration, intensity or spatial extents between the paired catchments (Alila et al., 2009; Zégre et al., 2010). Drought occurs on larger spatial and temporal scales, but it can still be challenging to compare single events. Therefore, we focus on the changes in the average and maximum drought metrics over a longer time period rather than on specific drought events.

The drought metrics mentioned in Section 2.3 were obtained from the drought analysis applied to each catchment and then compared between the paired catchments. The difference between the drought metrics of the human-influenced catchment and those of the benchmark catchment (Fig. 2) was calculated using the following equation (Eq. 1):

$$\textit{Percentage change due to the human influence (\%)} = \left[\frac{(\underline{\textit{Human} - \textit{Benchmark}})}{\textit{Benchmark}}\right] * 100 \qquad (1)$$

## 3. Case studies

We present results from two case studies to show the application of the paired-catchment approach to quantify the human influence on hydrological drought metrics. The case studies in the UK and Australia (Fig. 3) were chosen because of their contrasting climate, geology, and human activity. The UK case study is impacted by a water transfer scheme, has a temperate maritime climate and a relatively slow responding catchment, the Australian case study is impacted by groundwater abstraction, has a semi-arid climate and faster response times. These case studies were chosen based on the availability of data required for the assessment and analysis, including qualitative information on the human activities occurring in the catchments. For both human-influenced catchments we then applied the catchment selection scheme in Fig. 1 to find a suitable benchmark catchment.

Table 1 contains the main information about the two sets of paired catchments. For this application, we used observed discharge data (in mm/month). Precipitation data (mm/month) was used to check the similarity of average annual precipitation (within ± 10%) and distribution of monthly precipitation (Fig. 1, Table 1). For the UK case study, discharge time series and geology information were sourced from the National River Flow Archive (NRFA, 2018) and precipitation

data was obtained from CEH-GEAR (Keller et al., 2015). For the Australian case study, discharge and precipitation data and geology information were sourced from the Australian NSWs WaterInfo (NSW, 2018).

## 3.1 UK paired catchments: Blackwater and Chelmer

### 3.1.1 Paired-catchment selection

The Blackwater catchment receives water transfers as part of the Ely Ouse water transfer scheme for the greater London area (NRFA, 2018; Tijdeman et al., 2018). The scheme was introduced in 1972 by the Environment Agency to help address anticipated water stresses due to population increase and expansion and development in the South Essex area (AEDA, 1990). The Blackwater catchment was paired with a nearby catchment (Chelmer; Fig. 3) as its benchmark, due to their similarity in catchment characteristics (Table 1). Both catchments have a geology of mixed permeability superficial deposits (86-88%), a predominantly rural land use (Fig. 3) and similar annual rainfall totals (within 10%) (Table 1). Both catchments have very low urban extent (Chelmer 4.9% and Blackwater 5.4%; NRFA, 2018) and the land uses are very similar, with arable land covering 71% - 75% in both (NRFA, 2018; Fig. 3). There are no dams (GRanD database, Lehner et al., 2011) in either of the catchments. There are four sewage treatment works in the Blackwater catchment which have a minor impact on flow (NRFA, 2018), but there are no sewage treatment works in the benchmark catchment, the Chelmer. The observation data available for both catchments ran from 1972 to 2015 with no missing data, covering a number of important drought events in the UK.

### 3.1.2 Drought comparison results

The drought analysis shows that many droughts experienced in the natural catchment were alleviated in the human catchment due to the water transfer scheme (Fig, 4; Table 2). Notably, the 1976 UK drought was not as severe in the Blackwater catchment as its benchmark pair. A number of other major drought events occurred in Chelmer in the 1990s and 2003 were not seen in Blackwater, therefore showing that they were alleviated due to the elevated flows from the water transfer scheme (Fig. 4). The largest alleviation is seen in the total number of months in drought and the total deficit over the time series (Table 2), while the average duration and deficit decreased less due to the water transfer. This shows that the water transfer mainly reduced the number of droughts in the human-influenced catchment.

## 3.2 Australian paired catchments: Cox and Cockburn

### 3.2.1 Paired-catchment selection

The Cox catchment in south-eastern Australia has heavy groundwater abstraction for irrigation (Ivkovic et al., 2014). For the paired-catchment analysis, the benchmark catchment, Cockburn, was chosen based on its similarity in precipitation and geology (Table 1) and its proximity (Fig. 3). BFI for the benchmark catchment is much lower than that of the UK benchmark catchment, showing that the Australian catchments are responding faster to precipitation (Table 1). Observation data was available from 1982 – 2013, with no missing data.

The land use in both catchments is similar (see Fig. 3). Both catchments have a mix of natural savannahs and grassland used for grazing (Cox: 59%, Cockburn: 53%), with natural land cover composed of woody savannah (Cox: 32%, Cockburn: 28%) and forest (Cox: 6%, Cockburn: 13%). According to Green et al. (2011) the Cox catchment land use is "a combination of grazing and dryland cropping with agriculture". The benchmark catchment, Cockburn, is described as "the area is mainly

used for grazing with some dryland cropping and horticulture" (Green et al., 2011). The only difference between the catchments is the heavy groundwater abstraction in the Cox (Ivkovic et al., 2014), which is the human influence we are aiming to quantify. There are no dams (GRanD database, Lehner et al., 2011) or sewage treatment works, water transfers or other river regulations in either of the catchments.

*3.2.2 Drought comparison results*

Results showed an overall aggravation of drought metrics due to the groundwater abstraction in the human-influenced catchment, especially for the total number of months in drought and the total deficit (Fig. 5, Table 3). The well-documented Millennium Drought in Australia (2001 – 2009; Van Dijk et al., 2013) shows clearly as a series of hydrological drought events in the human-influenced catchment, whereas it was not as persistent in the benchmark catchment (Fig. 5). Like the water transfer in the UK case study, the water abstraction in Australia mainly influenced the drought frequency. Abstraction

resulted in more frequent drought events, not so much in more severe drought events. The maximum drought event characteristics are probably not affected (Table 3) because during these rare events the flow is zero in both catchments.

## 4. Discussion

As a scientific community we need to improve our understanding of the effect of human processes on hydrology and quantify the two-way interactions to be able to characterise, model and manage them (Srinivasan et al., 2017). These

processes can only be fully explored through observations (Jaramillo & Destouni, 2015). Here we have demonstrated that a paired-catchment approach is a suitable tool to assess and quantify the human influence on hydrological droughts using observation data. There are limitations to this approach, as any observation- and/or model-based approach has uncertainties. Model-based methods for quantifying the human influence have uncertainties associated to input data, parameters and model structure, which often does not include human processes (Wagener et al., 2004; Kreibich et al., 2017; Srinivasan et al. 2017),

and observation-based methods have uncertainties with regards to temporal or spatial resolution and data quality (McMillan et al, 2012; Coxon et al, 2015). Also, isolation of one type of human-influence is often more difficult in observation data. As discussed in this paper, for an effective paired-catchment analysis it is important for the catchment properties to be as similar as possible, enabling isolation of the human influence. In this study,  we selected the catchment pairs based on four characteristics: climate (precipitation, potential evapotranspiration), geology, land use, and qualitative information on human

influences (Fig. 1). Paired catchments in this study were chosen manually using expert judgement; however, recent advances in catchment classification and similarity frameworks (see Hrachowitz et al, 2013) could be an alternative because these

might provide a more objective and automated method to select paired catchments. Nevertheless, currently local knowledge is still a highly valuable part of catchment selection.

We focused on geology and precipitation as these two characteristics have been found elsewhere to be the most important drivers of hydrological droughts (Van Lanen et al., 2004a; 2013; Haslinger et al., 2014; Stoelzle et al., 2014; Van Loon & Laaha, 2015). In particular, we argue that differences in geology are more important in the analysis of low flows and droughts than for floods and annual flow analysis, because of the importance of catchment storage in the catchment response to precipitation and the drought propagation process (Van Lanen et al., 2013; Van Loon & Laaha, 2015; Barker et al., 2016). Differences in geology affect storage and response of catchments, and therefore response to meteorological drought events in one geology type could be very different to another, making it difficult to distinguish the effect of human influence from the effects of geological differences.

Similarity in precipitation is important as it is the driving meteorological factor for drought development. Both precipitation variability and annual averages should be similar for the paired catchments (Fig. 1). Firstly, it is important that the total precipitation inputs are similar for both catchments because of the cross use of the benchmark catchment threshold (Fig. 2). Significantly higher or lower precipitation in the benchmark catchment results in higher or lower discharge, which reduces the transferability of the threshold to the human-influenced catchment. We suggest a maximum difference of 10%. Alternatively, discharge could be scaled by dividing by average discharge, as done for example by Andréassian et al. (2012). Precipitation distribution may be very different even if total average values are similar, meaning that meteorological drought events experienced by the two catchments could be different. Ideally, the monthly and annual precipitation records of both catchments are also similar, which can be checked with the linear regression coefficient and $R^2$ values.

Moreover, it is important to make sure that there are not multiple human activities influencing streamflow in the human-influenced catchment. If we want to increase our understanding of the effect of different human activities on drought, we need to isolate the human influence under study. Here the focus was on the human activities of adding water via water transfer and removing water via groundwater abstraction and therefore we made sure that both catchments had the same urbanisation and that neither had dams or sewage treatment plants. These other human influences (land use change, water infrastructure) can, however, also be analysed using the paired-catchment approach.

Traditional paired-catchment experiments usually include a pre-disturbance period to assess the similarity of the catchments before treatment. Even when the catchments are subject to the same climatic variability, their hydrological response to an event may differ due to natural differences between the catchments (Folton et al., 2015), therefore a pre-disturbance period can possibly offer a quantification of these differences, e.g. through a regression equation (Hornbeck et al., 1993; Bari et al., 1996). However, generally this assumes a linear relationship between the catchments, which is a very crude assumption. Furthermore, a pre-disturbance period might not be available for paired catchments when they are not planned treatments and the human activity started before the hydrological measurements (Peñas et al., 2016). Peñas et al. (2016) tested the paired-catchment approach without pre-disturbance period (CI set up) to that with pre-disturbance period (BACI set up) and

found that over 80% of the impacted hydrological variables were identified correctly by the CI set up and, therefore, suggest to use this approach when no pre-disturbance data are available.

There are some challenges which remain specific to the paired-catchment analysis. Firstly, it can be very difficult to find a benchmark catchment, which is identical to the human-influenced catchment except for the human activity. Keeping a close proximity between the pairs can often help to reduce differences in geology and precipitation, however groundwater abstractions are known to impact the surrounding areas, beyond the catchment boundary, therefore neighbouring catchments might not be regarded as undisturbed by the human activity under study and should then not be used in a paired-catchment analysis. Instead, we suggest to compromise on another selection criterion (Fig. 1), such as precipitation, to locate a suitable benchmark catchment.

A second challenge, which is relevant for all observation-based methods, is data availability. Data availability and quality can severely affect the success of a catchment pairing or analysis. There is also a need for information on the type and extent of human disturbance, which may not always be available or known (Fig. 1). Differences in drought between the catchments cannot fully be attributed to the human influence, as there will always be a remaining uncertainty in the catchment pairing. However, the approach can be used for a first estimate of the human influence on drought, to steer campaigns to collect more data, and to complement other existing methods (e.g. model-based, large-sample and observation-modelling approaches).

Given the criticism of the method, the paired-catchment approach needs to be applied with care. For example, one-to-one comparison of events (chronological pairing) is not recommended (Alila et al., 2009; Zégre et al., 2010), instead multiple characteristics like frequency and magnitude taking into account all events in the period of record should be analysed to prevent underestimation of effects. Here, we transferred the threshold from the benchmark catchment to the human-influenced catchment to make sure that the effect of the human activities is not included in the threshold, again preventing underestimation (Rangecroft et al., 2019). However, there might be reasons to look at anomalies relative to each catchment's own threshold, for example by using standardised drought indices. This approach is often applied in paired-catchment studies focusing on flooding, in which the threshold for a peak-over-threshold analysis is determined for both catchments independently (e.g. Prosdocimi et al., 2015). A different way for calculating the threshold gives different outcomes (Rangecroft et al., 2019) and researchers need to be aware of this. It is important that it is clearly stated which approach is used and that benefits and limitations of that approach are discussed (Birkinshaw, 2014).

The application in this paper is the first use of paired catchments to quantify the human influence on hydrological droughts. One possible way forward from this could be to use data from published paired-catchment studies, but to focus the analysis on droughts rather than annual hydrological regime to assess how treatments of land cover change (e.g. deforestation, afforestation) have impacted hydrological droughts. These existing datasets have pre-disturbed time periods, and catchment selection has already been done rigorously. Another option would be to analyse other human activities such as urbanisation and the building of reservoirs.

## 5. Concluding remarks

In our human-dominated world we need to find ways to use our tools and methods to study the human influence on hydrology. In this study, we show the first application of the paired-catchment approach to quantify the human influence on hydrological droughts. We discussed how the selection of the paired catchments must be done with rigorous criteria (e.g. similar climate and geology), and identified the advantages and limitations of the approach. The main advantage is that automatically the same time periods are compared, therefore allowing climatic variability to be accounted for in the analysis. Furthermore, the approach uses catchment-scale observation data, allowing to gain information on the catchment level, with all of the catchment processes included. The other advantage of using an observation-only approach is that human actions and feedbacks are represented in the data, whereas most hydrological models currently do not include all anthropogenic processes yet. Whilst there are some uncertainties with regard to input data and catchment similarity, it is important to note that these uncertainties are similar to other methods used to quantifying the human influence on hydrological drought.

Here we showed how the paired-catchments approach, originally developed for treatment studies, and usually used for quantifying the impact of land use change on average discharge, could be used to look specifically at pre-existing and long-term human impact on hydrological droughts. We have used the method to analyse the impact of water transfer and groundwater abstraction in contrasting climate and geology settings. The example case studies in the UK and Australia clearly show an alleviation and aggravation of drought, respectively, due to the human activity compared to the benchmark catchment. However uncertainties remain in attributing these differences to the human influence under study, highlighting the importance of further analysis into how humans influence hydrological droughts. Paired catchments could be used to further investigate the impact of other human activities on hydrological droughts using observation data. Through an increased understanding of how human activities influence hydrological droughts, this knowledge can then be used for water resource management and for improving hydrological modelling.

## Author contributions

AVL and SR contributed equally to this work. AVL and HVL conceived the original idea, AVL developed the analysis code, SR performed the analysis, GC and FVO contributed local data for the case studies and helped with the interpretation of the results, all authors helped shape the research and discussed results. SR, AVL, GC, FVO, and JABN made the figures and AVL and SR wrote the manuscript with contributions from all co-authors.

## Acknowledgements

Anne Van Loon and Sally Rangecroft were supported by Rubicon NWO grant 'Adding the human dimension to drought' (reference number: 2004/08338/ALW). Gemma Coxon was supported by NERC MaRIUS: Managing the Risks, Impacts and

Uncertainties of droughts and water Scarcity, grant number NE/L010399/1. We thank Wouter Coomans (Wageningen University) for collating and analysing the Australian data. The research also contributed to the European Union funded H2020 project ANYWHERE (contract no. 700099). It is part of the IAHS Panta Rhei Drought in the Anthropocene working group and it supports the work of the UNESCO-IHP VIII FRIEND-Water programme and the programme of the Wageningen Institute for Environment and Climate Research (WIMEK-SENSE). Data and drought analysis R code are available upon request. We thank the editor, Markus Hrachowitz, and the reviewers Jamie Hannaford, Sivarajah Mylevaganam, and two anonymous reviewers for their comments and suggestions which greatly improved this paper.

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

**Table 1:** Catchment data of the paired catchments of both case studies

| Case study | Human activity | Catchment status | River/ Station | Catchment area (km$^2$) | Geology | Average annual precipitation (mm) | Average annual flow (mm) | BFI |
|---|---|---|---|---|---|---|---|---|
| UK | Water transfer scheme | Benchmark | Chelmer (37011 Churchend) | 72.6 | London clay and chalk, overlain with Boulder Clay | 591 | 91 | 0.43 |
| | | Human | Blackwater (37017 Stisted) | 139.2 | London clay and chalk, overlain with Boulder Clay | 579 | 194 | 0.5 |
| Australia | Ground water abstraction for irrigation | Benchmark | Cockburn (419016 Mulla Crossing station) | 907 | Alluvial overlying fractured rock (granite and sedimentary) | 665 | 64 | 0.24 |
| | | Human | Cox (419033 Tambar Springs station) | 1450 | Bedrock-contained alluvial valley | 732 | 21 | 0.21 |

**Table 2:** Paired-catchment results for UK, Blackwater (Human) and Chelmer (Benchmark).

| | Occurrence | Duration (months) | | | Deficit (mm) | | |
|---|---|---|---|---|---|---|---|
| | Frequency | Total duration | Average duration | Maximum duration | Total deficit | Average deficit | Maximum deficit |
| **Benchmark** | 22 | 86 | 3.9 | 10 | 163.8 | 7.4 | 29.6 |
| **Human** | 7 | 16 | 2.3 | 4 | 39.3 | 5.6 | 16 |
| **% difference due to the human influence** | -68% | -81% | -42% | -60% | -76% | -25% | -46% |

**Table 3:** Paired-catchment results for Australia, Cox (Human) and Cockburn (Benchmark).

| | Occurrence | Duration (months) | | | Deficit (mm) | | |
|---|---|---|---|---|---|---|---|
| | Frequency | Total duration | Average duration | Maximum duration | Total deficit | Average deficit | Maximum deficit |
| **Benchmark** | 14 | 52 | 3.7 | 10 | 14.3 | 1.0 | 3.4 |
| **Human** | 39 | 165 | 4.2 | 10 | 47.3 | 1.2 | 3.3 |
| **% difference due to the human influence** | +179% | +217% | +14% | 0% | +231% | +19% | -3% |

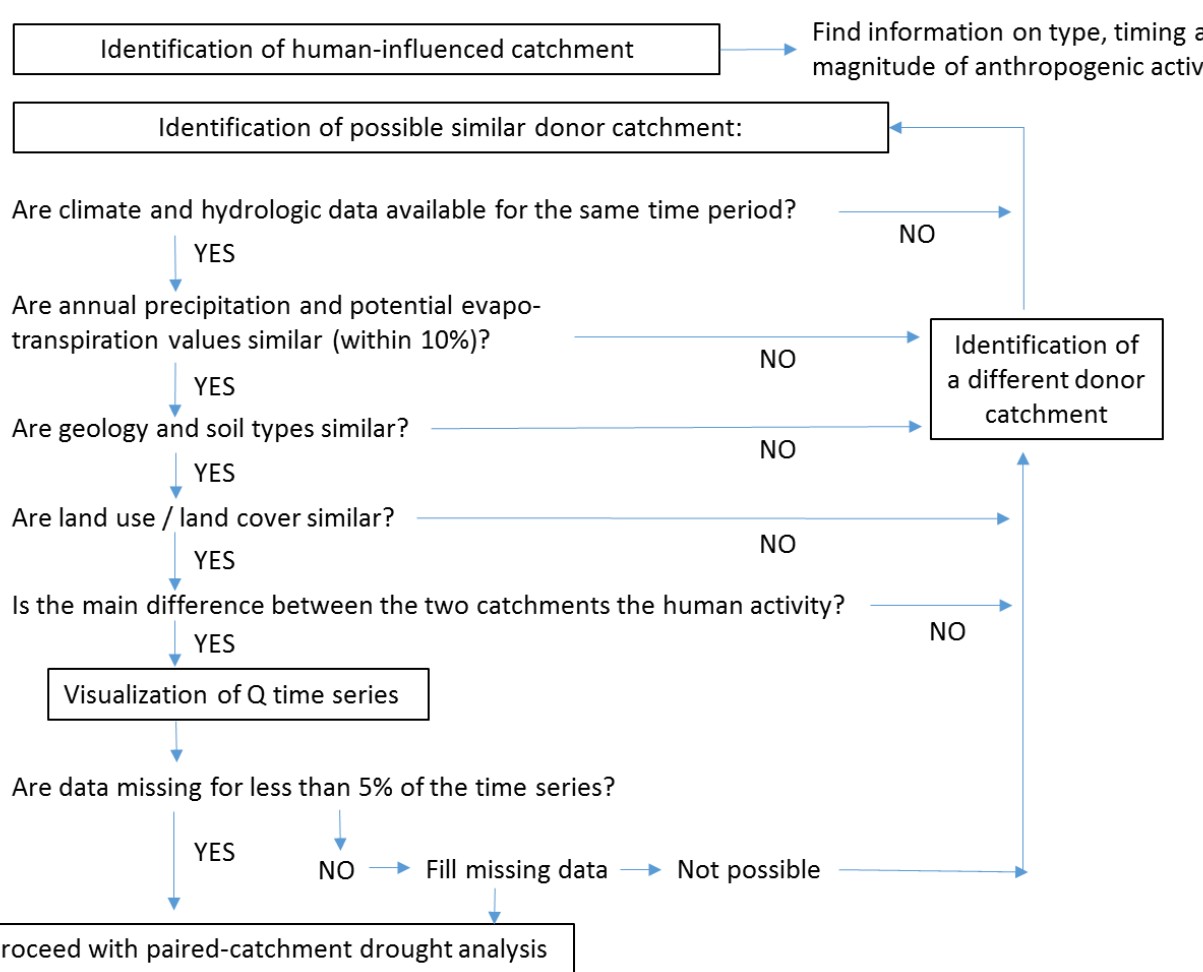

**Figure 1**: Flow diagram for choosing paired catchments for analysis of the human influence on hydrological drought.

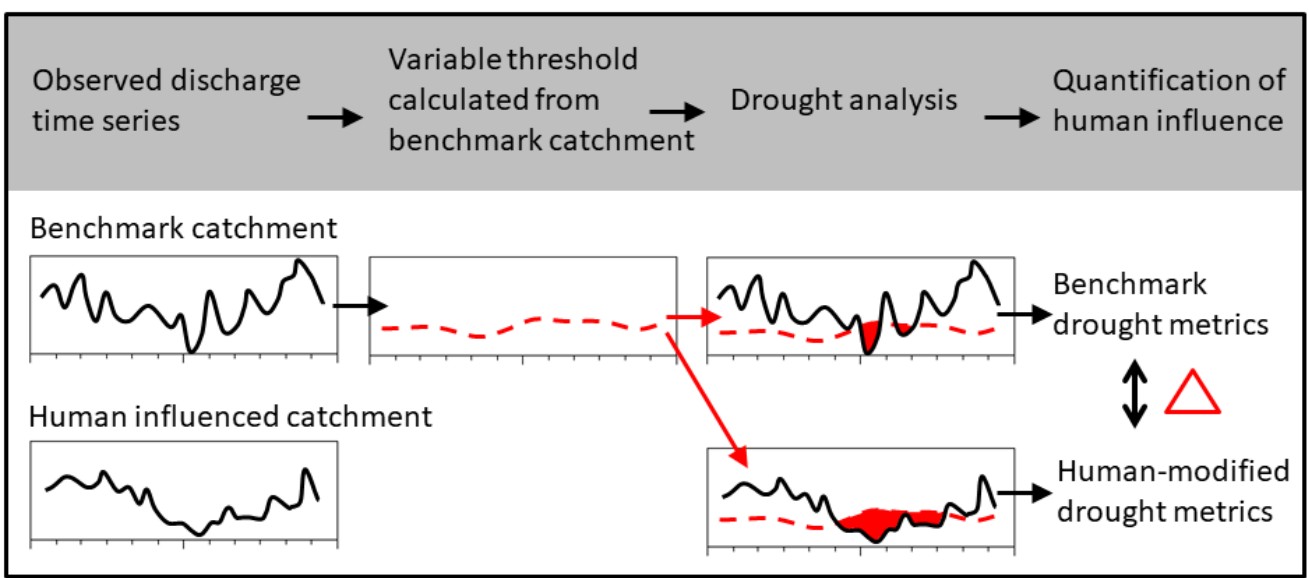

**Figure 2:** Diagram of the drought analysis method and quantification of the human influence on the hydrological drought metrics.

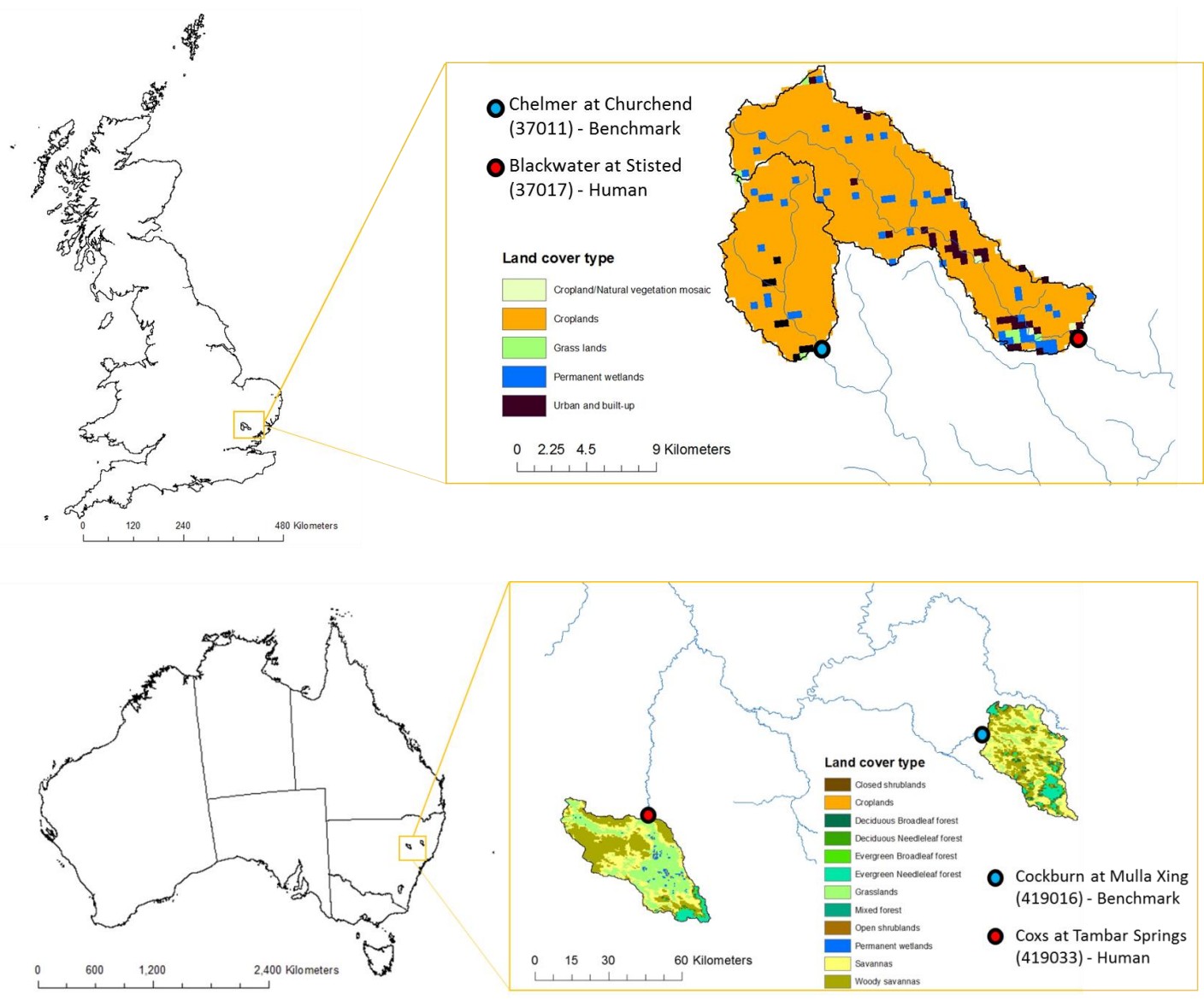

5    **Figure 3:** Paired-catchment case studies in the UK (top) and Australia (bottom), including the location of the gauging station and land use.

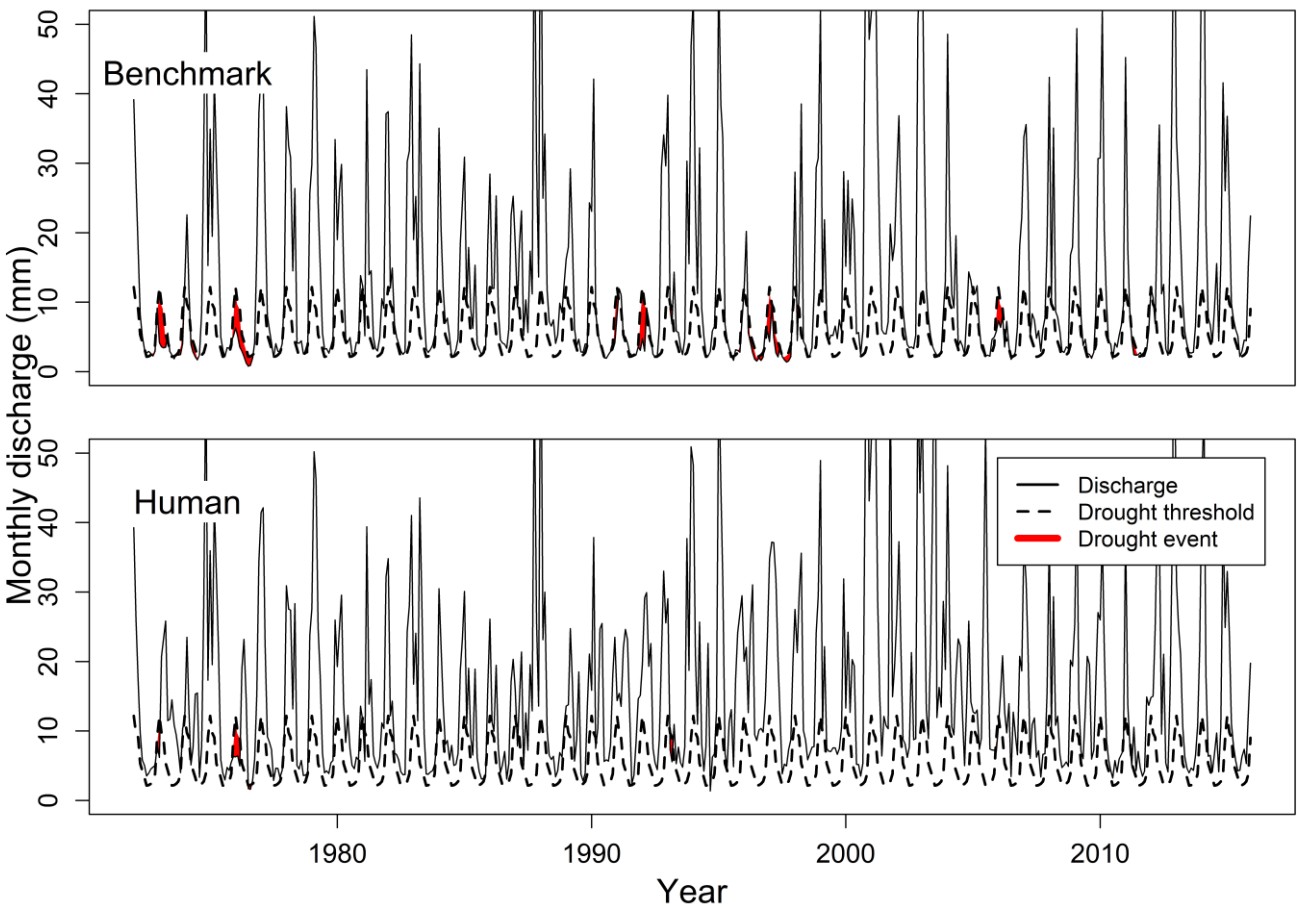

**Figure 4:** Drought analysis results for the UK pair, Blackwater (human) and Chelmer (benchmark) (1972 − 2015). Black solid line
represents streamflow, dashed black line represents Q80 variable threshold, and red areas are identified drought events.

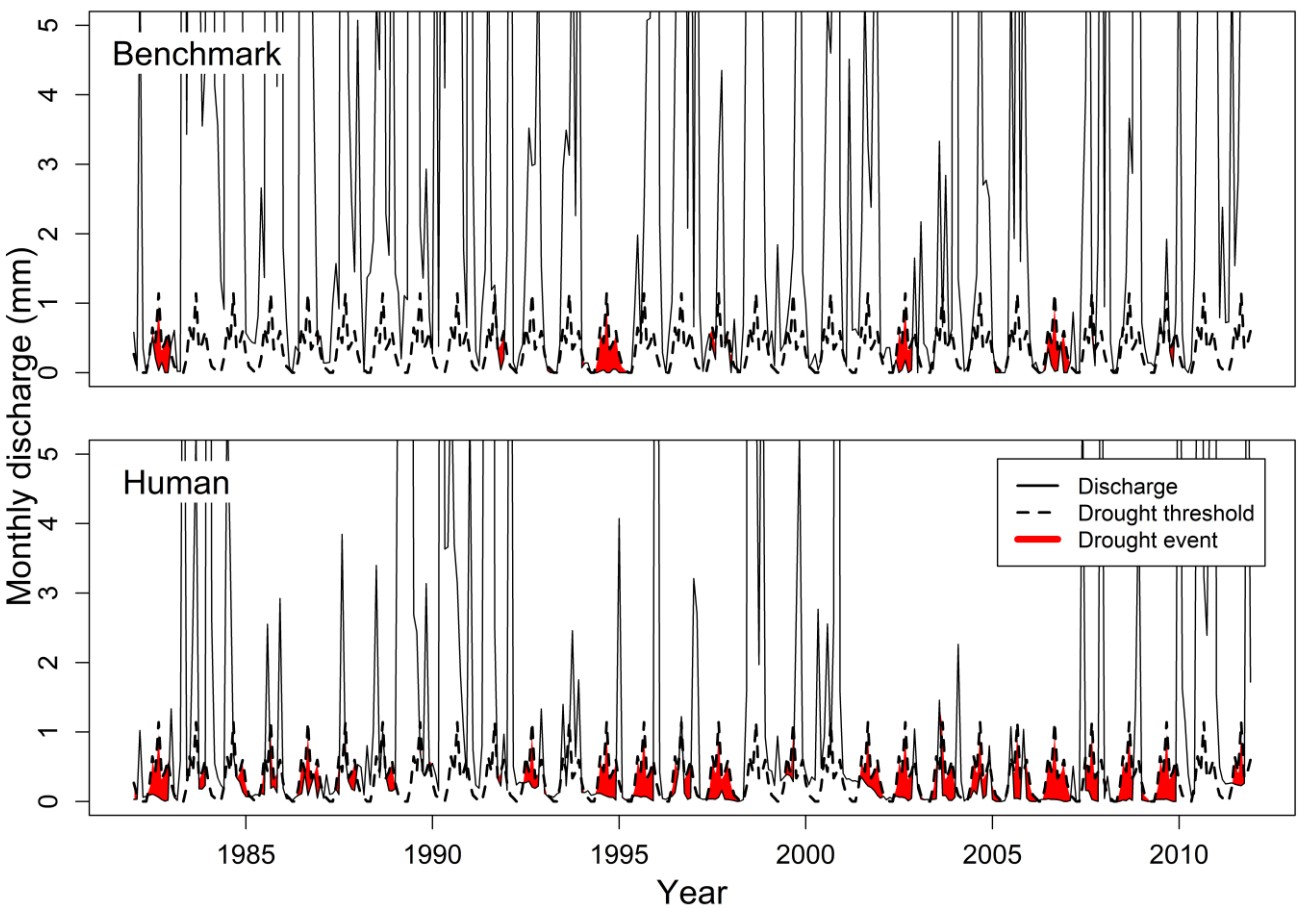

**Figure 5:** Drought analysis for the Australian pair, Cox (Human) and Cockburn (Benchmark) (1982 – 2013). Flow higher than 5 mm/month is not shown. Black solid line represents streamflow, dashed black line represents 80% variable threshold, and red areas are identified drought events.

