# Peer review of "Using paired catchments to quantify the human influence on hydrological droughts"

_Hydrology and Earth System Sciences, 2018_

## Short Comment (SC1) · 18 May 2018

S. Mylevaganam

mylevaganamsivarajah@gmail.com

The undesired consequences of extreme events (e.g., floods and droughts) in water sciences have forced many funding agencies and researchers to seek innovative methodologies to evaluate extreme events. In this manuscript, using the concept of paired catchments, the authors present an approach to evaluate drought conditions by means of drought metrics on drought frequency, drought duration, and deficit volume. The approach is demonstrated for a few selected catchments in UK and Australia.

1) As per the authors, research using the paired catchment approach to assess change in hydrological "droughts" due to land use and other human activities remains limited(see P-3 LN-29). However, as per the authors, research works using the paired

catchment approach to assess "low flows" due to land use and other human activities are found in the literature(see P-3 LN-24:29). What are low flows? What are droughts? I think, the distinction between low flows and droughts needs to be explicitly mentioned in the manuscript to assert the authors' statement that the research using the paired catchment approach to assess change in hydrological "droughts" due to land use and other human activities remains limited. Moreover, what is implied by "limited"? Should the authors include/cite the research works using the paired catchment approach to assess change in hydrological "droughts" due to land use and other human activities found in the literature?

2) As per the authors, drought analysis is normally conducted on the daily or monthly time step (see P-5 LN-23). Therefore, the authors use "monthly" data for the paired catchment analysis, even though the selected catchments are provided with data on daily time step? In the current version of the manuscript, the authors fail to state the reason for using "monthly" data for the paired catchment analysis. Is it the methodology (i.e., paired catchment) that is chosen in the proposed approach forces the authors to use the monthly data instead of daily data? Should the authors show the characteristics of the droughts within a month for the selected catchments? What is the minimum duration (in days) of the drought observed in the selected catchments? What is the minimum frequency (in days) of the drought observed in the selected catchments? Would not these details justify the applicability of selecting the monthly time step to conduct the drought analysis?

3) The use of paired catchment in the proposed approach is very much subjective. Is it possible to define hydrologically similar catchments without considering the landuse pattern and spatial orientation of the landuse? Would it be possible to completely define hydrologically similar catchments using precipitation, PET, and geology? The landuse pattern than the spatial orientation of the landuse may dramatically alter the flow pattern even for the same precipitation, PET, and geology?

4) In Table 3, for Dun (natural catchment), how did the authors compute the total number of months in drought and the frequency? Since the authors have analyzed from 1973 to 2013, there are 12*41(=492) monthly flow records. In other words, setting 80% as the threshold level yields around 98 months in drought (see Table 3). What is the definition of frequency? Is it meant for the return period of the drought? What is the unit of frequency?

5) In the current version of the manuscript, the authors evaluate the groundwater abstraction on the droughts. With groundwater abstraction, it is expected to have a depleted groundwater table. Consequently, as per Darcy's law, since one of the drivers (i.e., hydraulic gradient) is changed due to groundwater abstraction, a possible alteration on the base flow (see the reported BFI values in UK) that defines the low flow conditions in most of the rivers is expected. Would it be possible for the authors to show the rainfall pattern (daily) in the natural catchment and human altered catchment for some of the drought periods in UK?

6) As per the authors, the paired catchment approach has been a predominant method for detecting the effects of disturbance on catchment scale hydrology (see P-3 LN-14). To support this statement, the authors cite Zégre et al., 2010. Considering the fact that the paired catchment approach has been used for many decades (see P-3 LN-16), starting as early as 1920(see P-3 LN-19), would it possible for the authors to state the reason for citing Zégre et al., 2010 to support the statement(see P-3 LN-14).

7) Should the section 2.1 be re-written? The third paragraph of section 2.1 is about the method (i.e., paired catchment approach). In other words, the third paragraph of section 2.1 defines the method. However, the first and the second paragraphs of section 2.1 briefly summarize the crux of some of the previous research works using the paired catchment approach found in the literature. Does it make sense to mention the previous research works using the paired catchment approach at first and then define the method? From the reader's point of view, the third paragraph of section 2.1 should come first and then the previous research works using the paired catchment approach found in the literature.

8) The section that is devoted for discussion (i.e., section 4.0) is structured using some of the limitations of the approach presented in the manuscript. What is expected in the section (i.e., discussion) by a reader of the manuscript is left missing in the current version of the manuscript.

9) From the reader' point of view, the numbering of the sections is misleading. In section 2.2, the authors introduce the approach used in the current version of the manuscript to address the intended tasks (i.e., determination of drought metrics). However, in section 2.2, the authors ends the paragraph (see P-4 LN-17: here we outline the important elements for the "approach") to form the subsections that need to be listed under section 2.2. Should the sections 2.3, 2.4, and 2.5 be numbered as section 2.2.1, 2.2.2, and 2.2.3?

10) On P-3(see LN-22), what is meant by "see review of Brown et al., 2005"? Is it a review about Brown et al., 2015? On P-3(see LN-24), what is meant by "some studies included low flows"?

Minor Comments:

a) The authors' marriage to some of the words (e.g., "here" we suggest, "here" we outline, we "here" give, "here" we use, "here" the 80% percentile, "here" we have analyzed, "here" we present, "here" we have demonstrated, "here" we focused, "here" the focus was, and "here" we show) is a little off from what is expected in a scientific research paper. b) In Table 1, the title of the second column (i.e., "assessment for similarity") needs to be changed to reflect the cell values. c) In Table 2, the widths of the columns (e.g., column-7) need to be adjusted to fit the content. d) In Table 2, the gage numbers for the selected catchments in Australia are missing (see Figure 2 and Table 2). e) In Figure 3, the label for the x-axis should be "year"? f) In Figure 4, the label for the x-axis should be "year"?

---

## Referee Comment (RC1) · Anonymous Referee #1 · 21 May 2018

Rangecroft et al. aim to quantify the human influence on hydrological drought in two pairs of catchments in the United Kingdom and Australia. They use the paired catchment approach based on hydrological drought parameters such as drought duration, deficit volumes frequency, and total number of months of droughts and define hydrologic drought when monthly runoff is below its 20% percentile.

The article is well written, simple in the way that the methodology is straightforward and easy to replicate, the authors should be acknowledged for that. I have some suggestions to revamp the analysis, like a more thorough definition of the term "natural" and the inclusion of land and water-use characteristics in the basins, compulsory when performing a paired catchment analysis.

1. If the authors are trying to determine the presence of human influences on hydro-

logical drought, they should knowthe natural or reference influence to drought, i.e., the base hydroclimatic and basin characteristics. The authors do a rigorous hydroclimatic analysis in the way that their "paired" basins are same in size, close to each other, have same PET values, same P values. They have no similar geological characteristics but the authors are aware of this.

However, even though as the authors suggest, finding completely natural basins is difficult, their "isolation of the human influence" is coarse and undocumented, without properly accounting for differences of land use and water use between basins and without considering all the land use and water use already existing in the "natural" catchments.

Groundwater abstraction is not the only human effect that can influence drought characteristics. Just by looking in google earth at the Dun River upstream of Hungeford, UK, I see a basin that is completely agricultural, with barely any patch of natural vegetation. Since land cover/use is a control of water partitioning and runoff(Sterling et al., 2013), I don't know how "natural" is this basin. Additionally, by a quick search in google earth, I see that the river is also completely regulated by embankments and check dams/levies from beginning to end. Since flow regulation alters the partitioning of water and hence runoff (Jaramillo and Destouni, 2015), your reference conditions in terms of intra-annual variability and quantity of flow are already altered by regulation.

This means that your natural catchments are already affected and not eligible for a paired analysis for the purpose you need them. I would look harder for other basins with no water use/regulation and more pristine land covers to represent the "natural" condition, in the UK or abroad. Articles such as (Dynesius and Nilsson, 1994; Jaramillo and Destouni, 2015; Lehner et al., 2011; Nilsson et al., 2005) could help. The Authors can also choose instead of a "paired" analysis of catchments a paired analysis of two groups of catchments, this would make the analysis more robust.

Furthermore, the authors say that one of the basins suffers from groundwater abstractions while the other one doesn't, at least, for the UK pair. There is no quantification of this, how much, where, since when ground water abstraction in the human affected basin? Any evidence that in the other basin there is no groundwater abstraction, are you sure of this?

The authors should check this for the Australian case too.

2. Include some statistics of land use and water use in the Description of the area. Also, improve Figure 2, to show more information on each catchment.

3. The authors should put some statistical significance tests to support further their results.

So I like the idea, but a more thorough selection of catchments including land and water use conditions needs to be included in the study.

References

Dynesius, M. and Nilsson, C.: Fragmentation and Flow Regulation of River Systems in the Northern Third of the World, Science, 266(5186), 753–762, doi:10.1126/science.266.5186.753, 1994.

Jaramillo, F. and Destouni, G.: Local flow regulation and irrigation raise global human water consumption and footprint, Science, 350(6265), 1248–1251, doi:10.1126/science.aad1010, 2015.

Lehner, B., Liermann, C. R., Revenga, C., Vörösmarty, C., Fekete, B., Crouzet, P., Döll, P., Endejan, M., Frenken, K., Magome, J., Nilsson, C., Robertson, J. C., Rödel, R., Sindorf, N. and Wisser, D.: High-resolution mapping of the world's reservoirs and dams for sustainable river-flow management, Frontiers in Ecology and the Environment, 9(9), 494–502, doi:10.1890/100125, 2011.

Nilsson, C., Reidy, C. A., Dynesius, M. and Revenga, C.: Fragmentation and Flow Regulation of the World's Large River Systems, Science, 308(5720), 405–408,

doi:10.1126/science.1107887, 2005.

Sterling, S. M., Ducharne, A. and Polcher, J.: The impact of global land-cover change on the terrestrial water cycle, Nature Clim. Change, 3(4), 385–390, doi:10.1038/nclimate1690, 2013.

---

## Author Comment (AC1) · 21 Jun 2018

We thank you for your review and constructive comments. We hope that you find our responses satisfactory to the points that you make. We have included them as a supplementary pdf, where our responses to all of your points are in italic.

Thank you,

Sally Rangecroft (on behalf of the co-authors)

Please also note the supplement to this comment:
https://www.hydrol-earth-syst-sci-discuss.net/hess-2018-215/hess-2018-215-AC1-supplement.pdf

[Figure]

[Figure]

**Supplement:**

Dear Reviewer,

*We thank you for your review and constructive comments. We hope that you find our responses satisfactory to the points that you make. Our suggested response or actions are in italics.*

1a) If the authors are trying to determine the presence of human influences on hydrological drought, they should know the natural or reference influence to drought, i.e., the base hydroclimatic and basin characteristics. The authors do a rigorous hydroclimatic analysis in the way that their "paired" basins are same in size, close to each other, have same PET values, same P values. They have no similar geological characteristics but the authors are aware of this.

However, even though as the authors suggest, finding completely natural basins is difficult, their "isolation of the human influence" is coarse and undocumented, without properly accounting for differences of land use and water use between basins and without considering all the land use and water use already existing in the "natural" catchments.

Groundwater abstraction is not the only human effect that can influence drought characteristics. Just by looking in google earth at the Dun River upstream of Hungeford, UK, I see a basin that is completely agricultural, with barely any patch of natural vegetation. Since land cover/use is a control of water partitioning and runoff (Sterling et al., 2013), I don't know how "natural" is this basin. Additionally, by a quick search in google earth, I see that the river is also completely regulated by embankments and check dams/levies from beginning to end. Since flow regulation alters the partitioning of water and hence runoff (Jaramillo and Destouni, 2015), your reference conditions in terms of intra-annual variability and quantity of flow are already altered by regulation.

This means that your natural catchments are already affected and not eligible for a paired analysis for the purpose you need them. I would look harder for other basins with no water use/regulation and more pristine land covers to represent the "natural" condition, in the UK or abroad. Articles such as (Dynesius and Nilsson, 1994; Jaramillo and Destouni, 2015; Lehner et al., 2011; Nilsson et al., 2005) could help. The Authors can also choose instead of a "paired" analysis of catchments a paired analysis of two groups of catchments, this would make the analysis more robust.

*The reviewer makes an important comment about classifying the natural catchments as fully natural, and it has made us to reconsider the terminology used in our work to help represent this better. Whilst we fully agree with the reviewers comment, it is extremely difficult to find completely natural catchments that are similar enough to the influenced catchments. There are hardly any truly pristine catchments in the UK and most other regions across the world that are, headwater catchments, where there is no water use/regulation and pristine land cover. Furthermore, any pristine catchments are likely to be incomparable with the "human" catchments because of their different geology and climate. To address this, we propose to change the concept in our manuscript from a "natural" catchment, due to issues stated above, and instead refer to the catchment as "benchmark" to better represent this. We think that changing our terminology would strengthen the work introduced in this manuscript and show the representation and extent of human-influence better in a human-modified world.*

*In regards to the choice of the UK catchments – The Dun is one of the UK benchmark catchments (Harrigan et al., 2018), therefore it is considered as a "natural" catchment by the Centre for Hydrology and Ecology and we are confident in using it as a comparison pair. Crucially, the catchment does not need to be completely natural to be used in a paired catchment setting as long as you can isolate the influence you are interested in quantifying. Agriculture, land use and river regulation affects both catchments, so that if there is any influence this would be similar in both catchments and the comparison would still be valid. The main difference in the Kennet is the groundwater abstraction. Furthermore, in terms of the land cover/regulation on the River Dun, we argue that these factors have a minimal impact on low flows compared to climatic controls and water abstraction.*

*In regards to the choice of Australian catchments – we find that only the human-influenced catchment, Cox, has agriculture associated to it (Green et al., 2011): "The main land use in this region is a combination of grazing and dryland cropping with agriculture". Whereas, the benchmark catchment used, Cockburn, is described as "the area is mainly used for grazing with some dryland cropping and horticulture" (Green et al., 2011).*

*We propose adding some of these details about the catchments, their land uses and similarities into the revised paper to show this clearly, especially the change of terminology from "natural" catchment to "benchmark" catchment [information to be added to section 3.2].*

1b) Furthermore, the authors say that one of the basins suffers from groundwater abstractions while the other one doesn't, at least, for the UK pair. There is no quantification of this, how much, where, since when ground water abstraction in the human affected basin? Any evidence that in the other basin there is no groundwater abstraction, are you sure of this?

The authors should check this for the Australian case too.

*Unfortunately we do not have the exact numbers that can be shared for the groundwater abstractions (information protected by privacy laws).  For the UK, there have been many reports focusing on abstractions on the River Kennet and sustained pressure from local groups to reduce groundwater abstractions in the region. We will expanding on our first sentence in section 3.2 to include some of this information below.*

*Since the early 1990s Thames Water have abstracted water from the borehole at Axford, Wiltshire (close to Hungerford), to supply homes and businesses in Swindon and the Kennet valley (http://www.riverkennet.org/campaigns/abstraction-at-axford). Groundwater abstraction amounts are quoted to be 13,100 $m^3$/day at Axford (https://utilityweek.co.uk/thames-waters-river-kennet-abstraction-to-fall/). "Over-abstraction – this applies particularly to the Axford where the adverse impacts of abstraction have been proven, and a solution agreed… There have been other reviews of abstraction impacts in the Kennet Valley, and the impact of over-abstraction is a localised but important issue. Groundwater – the groundwater status in the catchment is poor. This is because there is not always enough groundwater to keep surface waters flowing" (Kennet Catchment Management Plan, 2012, p.5). "Water abstraction to meet the increased demand for water from*

*urban expansion and increased living standards has reduced the flow in the river." (Kennet Catchment Management Plan, 2012, p.14)*

*For Australia, the human influenced catchment, Cox, Ivkovic et al. (2014) have shown that the catchment is subject to heavy groundwater abstractions. Annual aquifer abstraction rates are quoted to be 11245 ML/yr (Barrett, 2012), whereas annual aquifer abstraction licenses for Cockburn are 4481 ML/yr (O'Rourke, 2010). Please note that definitions may differ slightly between the two reports: O'Rourke (2010) might be quoting entitlements, which may be higher than actual abstractions.*

2) Include some statistics of land use and water use in the Description of the area. Also, improve Figure 2, to show more information on each catchment.

*We will provide more information about land use in both paired catchments (e.g. information given in response to the first point).*

*Figure 2 we aimed to keep the information simple and clear, therefore Figure 2 was designed to show the location of the paired catchments and the station information. Whilst the UK has some detailed information about land use and water use, the same level of information is not available for Australian discharge stations. Furthermore, land use data is much harder to access and to include because of the usually higher spatial variability than precipitation, PET and geology. We can endeavour to look further into the possibility of visualising land use information on Figure 2; or instead we can overlay either topography, precipitation or geology in the revised version to help give the best visual of the catchments. We will also look to zoom in closer on the two Australian catchments in the Australian insert. We are reluctant to include any data from national datasets because the resolution of data is not detailed enough for the catchment scale. This said, we will include land use in our Table 1 to show that it is a consideration for catchment selection.*

3. The authors should put some statistical significance tests to support further their results.

*Whilst this would be a useful suggestion, unfortunately there are not enough data points in our analysis for statistical significance tests. We believe that quantifying the percentage change due to the human influence for the overall drought characteristics and showing the basic descriptive statistics of average, maximum and total duration and deficit volumes already provides very relevant information. We considered doing an event-by-event based analysis, but that proved to be impossible, partly because the input data is different for both catchments leading to variations between events. Thus, taking a more holistic look at the average, maximum and total drought duration and deficit over the entire time series is the most robust, and scientifically acceptable, information that we can provide. To perform statistical significance tests we would need more (replicate) catchment pairs, which is beyond the scope of this paper.*

So I like the idea, but a more thorough selection of catchments including land and water use conditions needs to be included in the study

*Thanks for your summary of the review. We hope that we sufficiently addressed this under items 1 and 2. We believe that adding this information will help to provide the right level of detail for the reader, and the change of terminology from "natural" catchment to "benchmark" catchment will help to show how this approach works in the Anthropocene.*

**References**

*Green, D., Petrovic, J., Moss, P. & Burrell, M., 2011. Water resources and management overview: Namoi catchment, Sydney: NSW Office of Water.*

*Harrigan, S., Hannaford, J., Muchan, K., Marsh, T. J. 2018. Designation and trend analysis of the updated UK Benchmark Network of river flow stations: the UKBN2 dataset. Hydrology Research, 49(2): 552-567*

*Kennet Catchment Management Plan. 2012. Kennet Catchment Partnership, [http://www.kennetcatchment.org/wp-content/uploads/2015/09/Kennet-Catchment-Plan-December-2012-revision-2.pdf](http://www.kennetcatchment.org/wp-content/uploads/2015/09/Kennet-Catchment-Plan-December-2012-revision-2.pdf)*

*NSW, G. 2018. Department of Primary Industries: Office of Water. [Online] Available at: [http://realtimedata.water.nsw.gov.au/water.stm](http://realtimedata.water.nsw.gov.au/water.stm)*

*O'Rourke, M. 2010. Peel Valley Catchment; Groundwater Status Report – 2010, NSW Office of Water, Sydney*

*Barret, C. 2010. Upper Namoi Groundwater Source - Status Report 2011, NSW Office of Water, Sydney*

---

## Author Comment (AC2) · 21 Jun 2018

Short comment by S Mylevaganam

*We thank you for your review and constructive comments. We hope that you find our responses satisfactory to the points that you make. Suggested response or actions are written in italics.*

1) As per the authors, research using the paired catchment approach to assess change in hydrological "droughts" due to land use and other human activities remains limited(see P-3 LN-29). However, as per the authors, research works using the paired catchment approach to assess "low flows" due to land use and other human activities are found in the literature(see P-3 LN-24:29). What are low flows? What are droughts? I think, the distinction between low flows and droughts needs to be explicitly mentioned in the manuscript to assert the authors' statement that the research using the paired catchment approach to assess change in hydrological "droughts" due to land use and other human activities remains limited. Moreover, what is implied by "limited"? Should the authors include/cite the research works using the paired catchment approach to assess change in hydrological "droughts" due to land use and other human activities found in the literature?

*We thank the reviewer for pointing out this for clarification. We define droughts within the paper at the end of the first paragraph (P2 LN8-10): "Here we use the definition of drought as a deficit in available water from 'normal' conditions (Wilhite & Glantz, 1985; Tallaksen & Van Lanen, 2004), with a focus on hydrological drought, which considers the deficit in streamflow." However we will include a definition of drought and low flows and add a brief description of how these two terms differ to make it clearer to the reader. We will use the definition by Smakhtin (2001): "minimum flow in a river during the dry periods of the year.... Low flows is a seasonal phenomenon, and an integral component of a flow regime of any river".*

*By "limited" we mean that there are actually no published articles specifically assessing effects of human activities on hydrological droughts using paired catchments in a rigorous manner. Tijdeman et al. (2018) use the concept of comparing droughts occurring in two similar catchments (e.g. Supplementary figure S4). They do not use a specific framework as we present here, but they explicitly state the potential use for it: "An alternative approach is based on the principles of paired catchment analysis, a concept that has been a foundation of process hydrology. Typically, a paired catchment study compares the flow regimes of nearby catchments with similar physical characteristics. The approach has been applied in numerous iconic experimental studies to investigate land use impacts on river flow (e.g. review of Brown et al., 2005). However, the paired catchment concept can also be used to study human influences on streamflow, using existing gauging station networks, if appropriate "donor" natural catchments with similar flow regimes can be found for "target" catchments with known influences (as conducted in the case of urbanisation effects on floods; Prosdocimi et al., 2015)." (Tijdeman et al., 2018, p.1053). We will change the statement in the manuscript to demonstrate that there are no published articles specifically assessing human activities on hydrological droughts using paired catchments.*

2) As per the authors, drought analysis is normally conducted on the daily or monthly time step (see P-5 LN-23). Therefore, the authors use "monthly" data for the paired catchment analysis, even though the selected catchments are provided with data on daily time step? In the current version of the manuscript, the authors fail to state the reason for using "monthly" data for the paired

catchment analysis. Is it the methodology (i.e., paired catchment) that is chosen in the proposed approach forces the authors to use the monthly data instead of daily data? Should the authors show the characteristics of the droughts within a month for the selected catchments? What is the minimum duration (in days) of the drought observed in the selected catchments? What is the minimum frequency (in days) of the drought observed in the selected catchments? Would not these details justify the applicability of selecting the monthly time step to conduct the drought analysis?

*As stated in the manuscript (p.5 L23), drought analysis is commonly done on the monthly time step, therefore it is deemed justifiable. Droughts are generally long phenomena with timescales of weeks to years, with drought-generating processes and associated impacts going beyond a daily time step. Therefore common time steps of weeks (few cases, and when used, a minimum of 15 days is often implemented), months, seasons, or multiple years. [P5 L22-25: "Although the paired catchment approach is most commonly used with annual data, it has also been used with monthly data (Bari et al., 1996; Brown et al., 2005). Drought analysis is normally conducted on the daily or monthly time step (e.g. Hisdal et al., 2004; Fleig et al., 2006; Van Loon & Van Lanen, 2012). Therefore, here we use monthly data for the paired catchment analysis."]*

*Furthermore, monthly data was used to avoid a need for pooling of drought events, and to allow minor droughts of <1 month to be dropped. This is stated in the Methods sub-section 2.5 Drought analysis P6 L6-7: "Drought events of only 1 month have been excluded from the analysis process so that drought events are longer than the time step of the threshold." We will look to state this earlier on in the Methods sub-section 2.4 Data requirements if this is deemed a more appropriate place to highlight this. Ultimately as long as the same time resolution and threshold is used for both catchments then the comparison remains valid.*

3) The use of paired catchment in the proposed approach is very much subjective. Is it possible to define hydrologically similar catchments without considering the landuse pattern and spatial orientation of the landuse? Would it be possible to completely define hydrologically similar catchments using precipitation, PET, and geology? The landuse pattern than the spatial orientation of the landuse may dramatically alter the flow pattern even for the same precipitation, PET, and geology?

*Land use data is much harder to access and to include because of the usually higher spatial variability than precipitation, PET and geology, therefore it could be a very difficult criterion to stick to. This said, we do consider land use (see reply to comment 2 of reviewer#1), we just do not include it as our fundamentals. We will indeed include this catchment characteristic into Table 1 to show the importance.*

*With regards to information about the land use of the pairs, for the UK, the Dun is one of the UK benchmark catchments (Harrigan et al., 2018), therefore it is considered as a "natural" catchment by the Centre for Hydrology and Ecology and we are confident in using it as a comparison pair. Land uses are similar for both catchments according to the NRFA.*

*With the Australian catchments – we find that only the human-influenced catchment, Cox, has agriculture associated to it (Green et al., 2011): "The main land use in this region is a combination of grazing and dryland cropping with agriculture". Whereas, the benchmark catchment used, Cockburn, is described as "the area is mainly used for grazing with some dryland cropping and horticulture" (Green et al., 2011).*

*This information will be included in the revised manuscript to show more information about land use.*

4) In Table 3, for Dun (natural catchment), how did the authors compute the total number of months in drought and the frequency? Since the authors have analyzed from 1973 to 2013, there are 12*41(=492) monthly flow records. In other words, setting 80% as the threshold level yields around 98 months in drought (see Table 3). What is the definition of frequency? Is it meant for the return period of the drought? What is the unit of frequency?

*Because minor drought events of <1 month have been dropped, it might result in slightly less drought events being identified as expected. However, our results show that the Dun has 93 months in drought, which is extremely close to this expected 98 months, and the differences seen between the two numbers is purely because of the dropping of minor drought events.*

*Frequency = number of drought events identified in the time period. This definition will be included in the revised version of the manuscript. It is uncommon to determine return periods and it is not a drought characteristic we choose to focus on. There is no unit for frequency as it is just a value, a number of drought events identified. We used the theory of runs to define events: see manuscript p. 5 Lines 27-28: "For each catchment, drought events and their metrics were identified with the commonly used drought analysis method, the threshold level method (Yevjevich, 1967; Tallaksen & Van Lanen, 2004; Van Loon, 2015)".*

5) In the current version of the manuscript, the authors evaluate the groundwater abstraction on the droughts. With groundwater abstraction, it is expected to have a depleted groundwater table. Consequently, as per Darcy's law, since one of the drivers (i.e., hydraulic gradient) is changed due to groundwater abstraction, a possible alteration on the base flow (see the reported BFI values in UK) that defines the low flow conditions in most of the rivers is expected. Would it be possible for the authors to show the rainfall pattern (daily) in the natural catchment and human altered catchment for some of the drought periods in UK?

*Indeed groundwater abstraction leads to lower groundwater heads, decreased groundwater gradients, lower groundwater inflow/base flow into the rivers, and hence more extreme drought in streamflow. We could indeed show some of the rainfall patterns for the drought periods if this would add to the paper. It is important to note that we do not take an event-by-event analysis though (as it is a challenge to pair drought events in different catchments), but look at overall averages, maximums and totals of drought duration and deficit over the entire time series to avoid this level of discretion.*

6) As per the authors, the paired catchment approach has been a predominant method for detecting the effects of disturbance on catchment scale hydrology (see P-3 LN-14). To support this statement, the authors cite Zégre et al., 2010. Considering the fact that the paired catchment approach has been used for many decades (see P-3 LN-16), starting as early as 1920 (see P-3 LN-19), would it possible for the authors to state the reason for citing Zégre et al., 2010 to support the statement(see P-3 LN-14)

*Thanks for this comment. In our revised version we will change this citation to one of an earlier time period, that of Bates (1921), which is the first paired-catchment study.*

7) Should the section 2.1 be re-written? The third paragraph of section 2.1 is about the method (i.e., paired catchment approach). In other words, the third paragraph of section 2.1 defines the method. However, the first and the second paragraphs of section 2.1 briefly summarize the crux of some of the previous research works using the paired catchment approach found in the literature. Does it make sense to mention the previous research works using the paired catchment approach at first and then define the method? From the reader's point of view, the third paragraph of section 2.1 should come first and then the previous research works using the paired catchment approach found in the literature.

*This restructuring of section 2.1 will be done in the revised version.*

8) The section that is devoted for discussion (i.e., section 4.0) is structured using some of the limitations of the approach presented in the manuscript. What is expected in the section (i.e., discussion) by a reader of the manuscript is left missing in the current version of the manuscript.

*The focus of the paper is the methodology and the two case study examples are just for illustrations (see L20-27 in Abstract: "Here we outline the methodological approach to quantifying this human influence on hydrological droughts and the requirements in catchment selection, as well as showcase the application using some example results from contrasting case studies in the UK and Australia with catchments heavily influenced by groundwater abstraction. Whilst the selection of the paired catchments must be done with rigorous criteria, this approach overcomes the impacts of climate variability in pre- and post-disturbance studies, and avoids assumptions considered when partly or fully relying on simulation modelling. We discuss important considerations for a successful analysis. This is the first application of this approach to quantify the human influence on hydrological droughts, demonstrating the use of this tool to study hydrology in our human-dominated world.")*

*We believe that because the aim of the paper is to introduce a simple but novel framework for quantifying the human influence on hydrological droughts, this is the important aspect to be discussed in this section, rather than the results of the two case studies which are used to demonstrate the method. Discussing the limitations and the possible ways forward for future research with this method is useful for future research, given its scalability to be performed in other locations.*

9) From the reader' point of view, the numbering of the sections is misleading. In section 2.2, the authors introduce the approach used in the current version of the manuscript to address the intended tasks (i.e., determination of drought metrics). However, in section 2.2, the authors ends the paragraph (see P-4 LN-17: here we outline the important elements for the "approach") to form the subsections that need to be listed under section 2.2. Should the sections 2.3, 2.4, and 2.5 be numbered as section 2.2.1, 2.2.2, and 2.2.3?

*We agree that the numbering here could be improved. We propose that section 2.1 Paired catchment analysis becomes its own section after the introduction. That way 2.2 onwards becomes the method and the relevant information for the analysis.*

*Alternatively, we can indeed change sections 2.3, 2.4 and 2.5 to be 2.2.1, 2.2.2 and 2.2.3.*

10) On P-3(see LN-22), what is meant by "see review of Brown et al., 2005"? Is it a review about Brown et al., 2015? On P-3(see LN-24), what is meant by "some studies included low flows"?

*We are refereeing to the published review of paired catchments written by Brown et al. (2005) titled 'A review of paired catchment studies for determining changes in water yield resulting from alterations in vegetation', however we can change the phrasing of this to be clearer: "see review paper of paired catchments by Brown et al., 2005"*

*Most paired catchment studies look at annual changes following treatment, and often focus on water yield and high flows, not on low flows or streamflow during droughts. Therefore we specifically point out that some look at low flows, but not the majority of the published literature. P3 L24-30 is an entire paragraph which summarises and points to the work which does use paired catchments to look at low flows: "In the 1990s, some studies included low flows (e.g. Keppeler & Ziemer, 1990; Scott & Smith, 1997). It has been found that clearcut harvesting can lead to an increase in low flows (Keppeler & Ziemer, 1990), while land cover conversion from grasslands, shrublands and croplands to forests can cause a decrease of low flows (Scott & Smith, 1997; Farley et al., 2005). Previous low flow works also suggested that in watersheds located in dry regions streams were likely to completely dry up following afforestation, and that the streamflow regime in those watersheds would change from perennial to intermittent (Farley et al., 2005; Jackson et al., 2009). However research using the paired catchment approach to assess change in hydrological droughts due to land use and other human activities remains limited."*

Minor Comments:

a) The authors' marriage to some of the words (e.g., "here" we suggest, "here" we outline, we "here" give, "here" we use, "here" the 80% percentile, "here" we have analyzed, "here" we present, "here" we have demonstrated, "here" we focused, "here" the focus was, and "here" we show) is a little off from what is expected in a scientific research paper.

*In our revised version we will endeavour to address this repetition.*

b) In Table 1, the title of the second column (i.e., "assessment for similarity") needs to be changed to reflect the cell values.

*This will be changed to 'Metric assessed'.*

c) In Table 2, the widths of the columns (e.g., column-7) need to be adjusted to fit the content.

*This will be rotated landscape and column widths expanded to fit the content.*

d) In Table 2, the gage numbers for the selected catchments in Australia are missing (see Figure 2 and Table 2).

*Thank you for pointing this out, these will be added to the revised version.*

e) In Figure 3, the label for the x-axis should be "year"?

*X axis label will be changed to Year rather than Dates in the revised version.*

f) In Figure 4, the label for the x-axis should be "year"?

*X axis label will be changed to Year rather than Dates in the revised version.*

*References*

*Bates, C. G. 1921. First results in the streamflow experiment, Wagon Wheel Gap, Colorado. Journal of Forestry, 19(4): 402-408.*

*Green, D., Petrovic, J., Moss, P. & Burrell, M., 2011. Water resources and management overview: Namoi catchment, Sydney: NSW Office of Water.*

*Smakhtin, V. U. 2001. Low flow hydrology: a review. Journal of Hydrology, 240: 147-186.*

---

## Referee Comment (RC2) · J. Hannaford (Referee) · 5 Jul 2018

This is a conceptual paper which aims to set out a methodology for characterising the human influence on streamflow drought via paired catchment analysis. Quite a substantial part of the paper reads as more of an extended perspective/opinion piece, but articulates a new method and applies it to two example catchments in UK and Australia.

This is a generally well written paper on an important topic and the paper has significant potential to make a very worthwhile contribution to the 'drought in the Anthropocene' debate. Paired catchment analysis is a staple of experimental hydrology, but is much harder to do in 'real world' examples when it is not possible to control all variables

except the main intervention of interest, and even the latter may be poorly understood. This paper tries to establish a method to allow this to be done even when there are no pre-disturbance periods, and without recourse to models. In this regard, it is potentially a very useful advance arising from a simple yet potentially very effective idea.

However, at present I don't think it can deliver on this promise as the methodology seems to have a flaw which I have outlined below in some detail. The method is predicated on the similarity of the donor natural catchment/target influenced catchment, but in the UK example at least, the catchments are not similar enough, very likely leading to over-estimation of the anthropogenic effect. I feel major revisions are needed to convincingly demonstrate the method, through modifying it to allow for some tolerance in the donor/target relationship, verifying the method using independent abstraction data, or benchmarking it against other methods.

Major Comment: impact of catchment (dis)similarity on the proposed method

There is a reason that paired catchment analysis using existing, gauged catchments is hard, and is rarely published (for drought or other topics): it is difficult to find suitable pairs. Even when catchments are in principle very similar (geology, rainfall etc), the concept of 'uniqueness of place' (as discussed at length in various papers by Keith Beven) is a major obstacle.

Despite this obstacle, data transfer is still possible as evident from the abundance of regionalisation methodologies available (a prior PUB decade, indeed!). However, transferring a threshold directly from one catchment (reading off the Q80 flow value from the natural catchment and applying it to the influenced one) to another seems like a potentially dangerous business.

Regionalisation methods use FDC statistics (e.g. transferring Q95 from donor to target catchments) for scaling purposes, but they tend to have lots of mechanisms built in to accommodate the fact that catchment similarity is imperfect: e.g. they are supported by multivariate regression based on catchment characteristics, and/or benefit

from pooling groups which place less emphasis on the relationship between any one donor and the target site. Finally, the uncertainties in regionalisation techniques are widely acknowledged. This might not be a problem if one is just trying to estimate flows at an ungauged site, and can report uncertainties; but in the present method any biases arising from the data transfer could be very misleading.

Put simply, the method applied in this paper can only work if the donor's natural flow regimes is near-identical to the 'theoretical natural' flow regime of the target site (i.e. what the regime would be in the 'world that might have been' with no human interventions). Any deviation between these regimes will be interpreted as anthropogenic; when it could just be due to variations between two catchments that appear quite similar but are in fact different.

This becomes problematic when one looks in detail at a catchment pair. I have investigated this for the UK pair as I am more familiar with UK hydrology, and have not commented on the Australian example. The Dun and Kennet are very similar indeed in terms of rainfall and geology, and make a good starting choice of study catchments. However, we can still see that the flow regimes are quite different. See the attached graph showing the two series scaled as runoff in mm, to account for the different catchment areas, as in the paper. The Kennet has a greater range in flows, with higher high flows and lower low flows (notwithstanding the abstraction effect). The Dun is more muted. The catchments are different in terms of runoff response/catchment function, despite their similar rainfall.

I'm not entirely sure of why the two catchments differ in terms of response, but it is likely that hydrogeology is a major factor. As acknowledged by the authors, in such chalk dominated catchments, the size/nature of the contributing catchment can be very different to the topographic catchment. Moreover, the geology of the chalk is heterogeneous and very complex.

The different regimes will have a significant bearing on the derived Q80. The net result

is (likely) that the natural Q80 transferred from the Dun is a biased estimator of the 'theoretical natural' flow of the Kennett. Given the more limited range of the Dun flows, my guess is the Q80 of the Dun will be somewhat higher, leading to inflated values for the deviations that are used to infer aggravated drought due to human effects.

As a result, I do not think the authors can claim 'attribution', and the claims of the paper need to be reconsidered. Even if the catchments were very close matches hydrologically, this would be 'weak' attribution. The method can be useful as a screening approach, but there remains a need to seek information on the influences in order to fully attribute. Note that this is an important difference in the urbanisation paper (Prosdocimi et al. 2015) or in the classic experimental catchments, which all incorporate some data on the intervention in question into the analysis. (e.g. the land cover data used by Prosdocimi et al.). The Kennett is very well known to experience major abstractions, which have been non-stationary over the series. But more could be done to follow this study up – there is anecdotal information on abstractions in various grey literature sources I found online (below).

In terms of results, the figures quoted in Table 3 seem very large. Given the nature of the debate around abstraction impacts, these figures could be quite contentious, as the Kennet is something of a poster child in the debate around sustainable abstraction, and the authors should do more to ensure they are meaningful. I'm not sure the method gives me enough confidence to get behind these figures. Other work suggests impacts in summer low flows of 10% - 40% (in major droughts). In addition, this paper uses the Kennet at Marlborough which is upstream of the single biggest abstraction at Axford which I imagine features in the 10 – 40% statistics quoted elsewhere.

(e.g. http://assets.wwf.org.uk/downloads/riverside_tales.pdf) http://assets.wwf.org.uk/downloads/case_study_kennet_final.pdf

Given these concerns, the authors could consider some approaches to bolster the method and provide verification – e.g., how this method performs relative to other ap-
proaches or modifications, e.g. detecting deviations based on rainfall (as in Tijdeman et al. 2018) and PE. In general the approach could be strengthened considerably by taking a more water balance approach, as done in the classic paired experimental catchment studies, and also in the study of Prosdocimi which incorporates climate variables to account for any confounding effects. But I'm not sure this would help as I think the differences in catchment function are possibly hydrogeological; this could be explored in more detail.

Finally, to really demonstrate the success of the method, it would be nice to have some independent verification of the suggested impacts. I appreciate access of abstraction data is not straightforward for the UK, but might be possible for one catchment, at least for derived data on impacts rather than particular abstractions. I would suggest some dialogue with the Environment Agency would be worthwhile, as there seem to be naturalised data (by decomposition and/or modelling) available for the Kennett for various past studies. http://www.mariusdroughtproject.org/wp-content/uploads/2017/01/MaRIUS_Kennet_ECG_Report_Jan2017.pdf

Specific Comments

A technical matter: In Figure 3, I'm surprised to see so little of the flows being below the threshold. It does not look like 20% of the flows are below the threshold to me – can the authors please check?

P2, L21. Another approach is using deviations in the P-Q relationship, e.g. Tijdeman et al. 2018.

P2, Intro. The paper would do well to refer to the expansive literature in hydroecology which also tackles a similar problem of estimating 'natural' flows for sites, against which impacted flows can be compared. The classic papers of Brian Richter are a good start, and I'm fairly sure methods have been proposed to transfer natural flow percentiles (but using a whole FDC approach; try the DHRAM work by Andrew Black, Dundee as a start). Another area where this is done routinely is through the LowFlows software

product, a regionalisation product which estimates natural and disturbed FDCs ar any site. It's not drought specific, but definitely has a very similar aim.

P4, Sect. 2.3. Given the concerns raised about the UK catchments, this section needs to be reconsidered.

P6, L2. The 80th percentile is not what is being used here. This paper uses the 20th percentile, or, as is most commonly referred to in hydrology, Q80: the 80% non-exceedance threshold from the flow duration curve.

Discussion: is generally very insightful but definitely needs reconsidering in light of catchment selection issues, and claims about attribution need to be moderated.

[Figure]

**Fig. 1.**

---

## Author Comment (AC3) · 13 Aug 2018

**Authors reply to Reviewer 2's comments**

Please find our responses and actions to Reviewer 2's comments below. The reviewer's comments are in italic and labelled with a (1) and our response is in normal font, labelled with a (2). Frequently we have placed a number of the reviewer's comments under the same section if they are related and our response covers them. Figures and tables shown here only in this reply have the prefix R (e.g. Figure R1) to distinguish them from those shown in the manuscript. In our existing responses to other HESSD reviewers, we have suggested that we rename the "natural" catchment to be termed the "benchmark" catchment; therefore, the new term is used throughout in our response here.

There are a few major things that we will change in the manuscript in general, which we would like to highlight here before discussing the specific reviewer's comments. We bullet point these major changes here, which all relate to the framing of the paper and the content:

- Paired catchment analysis as a complementary method. We now state clearly when introducing the method that the paired catchment analysis is a complementary method to help gain insight into the human influence in a catchment.
- Methodological framework flow diagram to follow. As this is a methods paper, we keep the focus on the methodology aspects and not the results of the case studies. We will introduce a flow diagram to illustrate more clearly the pairing of catchments.
- Demonstrating the method with case studies of different human activities, which alleviate and aggravate droughts. We now display the application of the paired catchment analysis on different human activities by showing a case study in which human activities alleviate droughts (UK Blackwater) as well as the existing Australian case study, which shows an aggravation of droughts due to the human activity. This changing of the UK paired catchment case study from the Kennet to the new case study of Blackwater also addresses a number of Reviewer 2's concerns about the UK pairing, as well as showing an alleviating human activity. The other reviewers also picked up this dissimilarity of the Kennet pairing.

Reply to Reviewer's comments

*(1) This is a generally well written paper on an important topic and the paper has significant potential to make a very worthwhile contribution to the 'drought in the Anthropocene' debate. It is potentially a very useful advance arising from a simple yet potentially very effective idea.*

(2) We thank the reviewer for their overall view of the importance of the topic and paper.

***(1) Major Reviewer's comment: pairing of UK catchment***

*(1) Major Comment: impact of catchment (dis)similarity on the proposed method. The method is predicated on the similarity of the donor natural catchment/target influenced catchment, but in the UK example at least, the catchments are not similar enough, very likely leading to over-estimation of the anthropogenic effect.*

*(1) Catchments are different in terms of runoff response/catchment function, despite their similar rainfall.*

*(1) Given the more limited range of the Dun flows, my guess is the Q80 of the Dun will be somewhat higher, leading to inflated values for the deviations that are used to infer aggravated drought due to human effects.*

(2) We realise that a large proportion of the Reviewer 2's comments are addressing the pairing of the UK case study specifically, highlighting the issue of the (dis)similarity between the UK pair. These comments help to show the level of information needed for the proposed paired-catchment method, and the importance of a suitable pairing. Although the original catchments of the Kennet and the Dun are similar in terms of their rainfall and geology, we agree with the reviewer that the catchments are different in terms of runoff response and it is difficult to attribute this difference solely due to human influences without further information on abstraction data. We have taken on board some of the reviewer's comments to help improve the pairing.

As this is a methods paper and the case studies shown purely demonstrate the method, we found a UK pair that is more similar, which is demonstrated by other studies (e.g. Tijdeman et al., 2018) and which helps to demonstrate the use of the paired catchment analysis on a different human activity. Therefore, we have changed our UK case study from the Kennet to the Blackwater catchment (Figure R1). The Blackwater catchment has a water transfer scheme delivering excess water to the Essex area, helping to keep it wet during the summer in dry years (Robinson, 2011). Tijdeman et al. (2018) suggest the pairing with a similar catchment, Chelmer, to represent the natural situation with very similar catchment characteristics. Chelmer is identified to have minor artificial influences, with the main difference being the human activities present in Blackwater.

[Figure]

**Figure R1:** Discharge plotted (mm/month) for both catchments in new UK case study pair: Benchmark, Chelmer (blue) and human-influenced, Blackwater (red)

While the focus of this methods paper is to use the case studies to illustrate the method rather than attributing differences to specific causes, we think that choosing a different and less controversial pairing with more similar runoff response will better highlight the advantages of the method.

Please find below the rewritten section to replace "Section 3.2 UK paired catchments: Kennet and Dun" for the paper and the new case study and results for Table 3 and Figure 3, and updated Table 2:

"**Section 3.2 UK paired catchments: Blackwater and Chelmer**

The Blackwater catchment receives water transfers as part of the Ely Ouse water transfer scheme for the greater London area (NRFA, 2018; Tijdeman et al., 2018). The scheme was introduced in 1972 by the Environment Agency to help address anticipated water stresses due to population increase and expansion and development in the South Essex area (AEDA, 1990). Blackwater catchment was paired with a catchment nearby as its benchmark pair, Chelmer (Figure 2), due to its similarity in catchment characteristics (Table 2). Both catchments have mixed permeability superficial deposits geology (86-88%), are predominantly rural land use catchments and have similar annual rainfall totals (within 10%) (Table 2). Both catchments have very low urban extent (Chelmer 4.9% and Blackwater 5.4%) and the land uses are very similar, with arable land covering ~70-75% in both (NRFA, 2018). The observation data available for both catchments ran from 1972 to 2015 with no missing data, covering a number of important drought events in the UK.

The drought analysis shows that many droughts experienced in the natural catchment were alleviated in the human catchment due to the water transfer scheme (Figure 3; Table 3). Notably, the 1976 UK drought was not as severe in the Blackwater catchment as its benchmark pair. A number of other major drought events occurred in Chelmer in the 1990's and 2003 were not seen in Blackwater, therefore showing that they were alleviated due to the elevated flows from the water transfer scheme (Figure 3)."

**Table 3:** Paired catchment analysis results for UK case study, Blackwater (Human) and Chelmer (Benchmark).

| | Occurrence | Duration | | Deficit | | |
|---|---|---|---|---|---|---|
| | Frequency | Total no. of months in drought | Average duration (months) | Maximum duration (months) | Total deficit (mm) | Average deficit (mm) | Maximum deficit (mm) |
| **Benchmark** | 22 | 86 | 3.9 | 10 | 163.8 | 7.4 | 29.6 |
| **Human** | 7 | 16 | 2.3 | 4 | 39.3 | 5.6 | 16 |
| **% increase due to the human influence** | -68% | -81% | -42% | -60% | -76% | -25% | -46% |

[Figure]

**Figure 3:** Drought analysis results for the UK pair, Blackwater (human) and Chelmer (benchmark) (1972 – 2015). Black solid line represents streamflow, dashed black line represents Q80 variable threshold, and red areas identified drought events.

*Table 2: Catchment data about each paired catchment case study*

| Case study | Human activity | Catchment status | River/ Station | Catchment area (km²) | Geology | Average annual precipitation (mm) | Average annual flow (mm) | BFI |
|---|---|---|---|---|---|---|---|---|
| **UK** | Water transfer scheme | Benchmark | Chelmer (37011 Churchend) | 72.6 | London clay and chalk, overlain with Boulder Clay | 591 | 91 | 0.43 |
| | | Human | Blackwater (37017 Stisted) | 139.2 | London clay and chalk, overlain with Boulder Clay | 579 | 194 | 0.5 |
| **Australia** | Ground water abstraction for irrigation | Benchmark | Cockburn (419016 Mulla Xing station) | 907 | Alluvial overlying fractured rock (granite and sedimentary) | 665 | 64 | 0.24 |
| | | Human | Cox (419033 Tambar Springs station) | 1450 | Bedrock-contained alluvial valley | 732 | 21 | 0.21 |

By changing the UK case study, we hope to have answered the reviewer's questions. If needed, we could however also increase the number of case studies presented in the manuscript to fully demonstrate the method. In doing so, we would also cover a range of human activities, to showcase how the method can be used on different human activities and to take the emphasis away from the contentious results of only groundwater abstraction case studies. As well as a the UK Blackwater case study (water transfer) and Australia Cox case study (groundwater abstraction), we can show another UK case study (e.g. Candover, flow augmentation scheme) and a Mexican case study (e.g. Torres, urbanization). However, we have this multiple case study comparison currently as future

research in preparation for publication and we would prefer to keep this paired catchments paper as a methods based paper.

*(1) In terms of the results, the figures quoted in Table 3 seem very large. Given the nature of the debate around abstraction impacts, these figures could be quite contentious, as the Kennet is something of a poster child in the debate around sustainable abstraction, and the authors should do more to ensure they are meaningful.*

(2) We agree that the numbers were very large for our Kennet case study, but this might be realistic given the published concern over over-abstraction in the Kennet area, and recent changes to abstraction rates to reduce this (2014 EA issued a notice reducing the permitted abstraction from Axford and revoked the abstraction licence on the little River Og). Therefore it is legitimate that impacts seen over the previous decades could reflect this over-abstraction.

e.g. "Thames Water accused over dry River Kennet in Wiltshire", BBC, 27 September 2011 (https://www.bbc.co.uk/news/uk-england-wiltshire-15076406)

e.g.2: "For over 20 years there has been local concern that flows in the Upper Kennet above Marlborough have been affected by over-abstraction' (Kennet catchment partnership). (http://www.kennetcatchment.org/issues/over-abstraction/)

However, given that the focus of this paper is to introduce the method, there is concern that bringing in this level of information about the individual case studies will distract away from this. We will include a summary sentence demonstrating the level of required information about the human activity and reported impacts of the presented case studies, but a more detailed paragraph should be kept for Supplementary information (if needed) rather than to extend the current paper and take the focus away from the method itself.

(1) **Overall reviewer's comments: considering other approaches to bolster the method / convincingly demonstrate the method:**

*(1) Given these concerns, the authors could consider some approaches to bolster the method and provide verification – e.g., how this method performs relative to other approaches or modifications, e.g. detecting deviations based on rainfall (as in Tijdeman et al. 2018) and PE.*

*(1) Paired catchment analysis is a staple of experimental hydrology, but is much harder to do in 'real world' examples when it is not possible to control all variables except the main intervention of interest, and even the latter may be poorly understood.*

*(1) I feel major revisions are needed to convincingly demonstrate the method, through modifiying it to allow some tolerance in the donor/target relationship, verifying the methods using independent abstraction data, or benchmarking it against other methods.*

(2) We argue that the paired catchment approach can be extremely beneficial if you have the right level of information to justify the pairing, but not enough detailed data on the human influences

themselves (i.e. time series of abstraction rates) or to run alternative approaches (i.e. upstream-downstream approach, Rangecroft et al., 2016; observation-modelling approach, Van Loon and Van Lanen, 2013; 2015). Therefore, the paired catchment approach can help provide insight into the human influence in the catchment from the observation data available and might be a first step to obtain more data to apply alternative approaches. Exploring the use of other methods to obtain more robust results for the example case studies is not the purpose of the paper, but we will definitely reframe to explain that this method can be complementary to others.

Furthermore, we have also changed the framing of the paper in terms of the case studies presented. We now propose to show a case study which has a human activity aggravating hydrological droughts (e.g. Australia Cox – groundwater abstractions) but also a case study in which the human activity alleviates hydrological droughts (e.g. UK Blackwater – water transfer). We believe that this will help demonstrate the method and its applicability better.

To help illustrate the method further, we also propose the addition of a flow diagram for the pairing of catchments for the analysis.

*(1) In general the approach could be strengthened considerably by taking a more water balance approach as done in the classic paired experimental catchment studies, and also in the study of Prosodcimi which incorporates climate variables to account for any confounding effects.*

(2) In the new framing we will emphasise the use of the paired catchment approach when less detailed information is available to still make an assessment of the difference between the two catchments. The use of the same time period as a comparison between the two catchments means that overall climate is accounted for in the analysis, which is an advantage compared to methods comparing data pre- and post-disturbance.

The water balance approach requires a high level of information, including actual evapotranspiration, storage changes (e.g. soil moisture, groundwater), and does not focus on drought specifically; instead we are looking for a method which can be applied to multiple case studies to give an idea about the impact of human activity on drought only based on commonly available data.

*(1) Reviewers comment: include abstraction rates information*

*(1) verifying the method using independent abstraction data, or benchmarking it against other methods.*

*(1) The Kennett is very well known to experience major abstractions, which have been non-stationary over the series. But more could be done to follow this study up – there is anecdotal information on abstractions in various grey literature sources I found online (below).*

(2) As mentioned, some abstraction values are available for the UK case study through grey literature. However, it is difficult to obtain actual abstraction values, rather than licenced values, and

for the Australian case there is no information available at all. Therefore it can be very hard to give an estimate of the overall net groundwater abstraction during the time series.

We agree that human activities are non-stationary over the investigated time period. This is one of the reasons why we calculate overall drought characteristics for the whole time series, rather than an event-by-event analysis.

*(1) Reviewers comment: Impacts and abstraction data*

*(1) Finally, to really demonstrate the success of the method, it would be nice to have some independent verification of the suggested impacts. I appreciate access to abstraction data is not straightforward for the UK, but might be possible for one catchment, at least for derived data on impacts rather than particular abstractions.*

(2) We would like to be careful not to allocate too much of the paper to the two case studies as we feel that this will detract away from the focus of the paper, which is the method itself. However, we have included more information here to give a background on the new UK pairing (see new Section 3.2). It is also important to note that gaining the level of information we have for the UK is possible, but it is difficult to get abstraction data for many other regions of the world, including Australia.

*(1) I would suggest some dialogue with the EA would be worthwhile, as there seem to be naturalised data (be decomposition and/or modelling) available for the Kennet for various part studies.*

(2) We fully agree that a comparison between observation data and naturalised data for the human influenced catchment would complement this method and paper, to see if the results are similar. However, this is not the focus of the paper and we do not fully agree with the use of the EA naturalisation data as we do not have information about which processes and activities are included in the modelling and/or decomposition. The naturalisation data has large uncertainties as well and cannot be used as a benchmark or independent verification.

*(1) Reviewers comment: using threshold from benchmark catchment*

*(1) However, transferring a threshold directly from one catchment (reading off the Q80 flow value from the natural catchment and applying it to the influenced one) to another seems like a potentially dangerous business. This might not be a problem if one is just trying to estimate flows at an ungauged site, and can report uncertainties; but in the present method any biases arising from the data transfer could be very misleading.*

(2) The transferring of thresholds from the donor catchment to the influenced catchment is a basic principle of the paired catchment approach. Furthermore, the transferring of thresholds from the benchmark situation to the human-influenced situation has been used in other existing literature, with regards to the comparison of naturalised data and observed flow (observation-modelling framework, Van Loon & Van Lanen, 2013; Van Loon & Van Lanen, 2015). It is also used by the largescale hydrological modelling community when analysing future droughts – use of pristine threshold for both the pristine and human scenarios to calculate the modelled human impact on hydrological droughts (e.g. Wanders & Wada, 2015).

The main issue with using a threshold established on the human-influenced catchment discharge is that the effect of the human activities is then included in the threshold used to calculate droughts. This makes it harder to compare the observed situation with the expected normal. The use of the benchmark catchment for the threshold allows a better representation of the expected normal without the human activity.

We currently choose not to explore this avenue of analysis because it would complicate the paper too much. However, we show the results here for one case study, UK Blackwater, to example the difference that can be observed for the reviewer (Figure R2 & Table R1). Because the human influence is now included in the threshold (Figure R2), using the own threshold results in lower numbers of the human influence on drought (Table R1). We feel that this underestimates the human influence on drought, but if required, we can include both the application of own thresholds and the application of the benchmark threshold in the manuscript.

**Figure R2:** Variable Q80 thresholds for the benchmark catchment (blue) and human catchment (red)

**Table R1: UK Blackwater case study:** Percentage change in hydrological drought characteristics due to human influence calculated with either own station thresholds or benchmark threshold.

| Hydrological droughts % change detected in human catchment compared to benchmark | Freq. | Duration | | | Deficit | | |
|---|---|---|---|---|---|---|---|
| | | Average | Max | Total | Average | Max | Total |
| Own threshold | +14 | -32 | -40 | -23 | -24 | +1 | -14 |
| Benchmark threshold | -68 | -42 | -60 | -81 | -25 | -46 | -76 |

*(1) Reviewers comment: Impact of catchment (dis)similarity on the proposed method*

*(1) Put simply, the method applied in this paper can only work if the donor's natural flow regimes is near-identical to the 'theoretical natural' flow regime of the target site. Any deviation between these regimes will be interpreted as anthropogenic; when it could just be due to the variations between two catchments that appear quite similar but are in fact different.*

*(1) It is difficult to find suitable pairs. Even when catchments are in principle very similar (geology, rainfall etc), the concept of 'uniqueness of place' is a major obstacle.*

(2) We agree that 100% proof of similarity cannot be given when datasets are used of catchments already disturbed by the human activity. Experimental hydrology (control and treated catchments/plots) and models are also unable to provide this. We can only reliably check the similarity of both catchments if we have long time series (preferably 30 years to calculate the thresholds) for the donor and the human-influenced catchment prior to human disturbance. In practice, this will be very rare or none-existing for catchment pairs.

*(1) As a result, I do not think the authors can claim 'attribution', and the claims of the paper need to be reconsidered.*

*Note that this an important difference in the urbanisation paper (Prosdocimi et al., 2015) or in the classic experimental catchments, which all incorporate some data on the intervention in question into the analysis (e.g. the land cover data used by Prosdocimi et al.).*

(2) We agree that we can rephrase our results from attribution to likely cause. We will cover this in our revised discussion section, stating that is it about all human influences (and some minor differences between catchments under natural conditions).

We can include abstraction rate information, even if from grey literature sources online, when available, and more detailed information on land use. As previously stated, we are cautious to not include too much detail about the two case studies (Cox, Blackwater) as it is a methods paper and the focus should remain on demonstrating the method, not on the discussing the case studies themselves.

***(1) Specific reviewer's comments***

*(1) A technical matter: In Figure 3, I'm surprised to see so little of the flows being below the threshold. It does not look like 20% of the flows are below the threshold to me – can the authors please check?*

(2) We can confirm that it is correct. The total number of months in drought (Table 3) is close to 20%: 93 months in drought out of 540 months (17%). It is not the full 20% because we have dropped minor droughts that were only 1 month in duration.

*(1) P2, L21. Another approach is using deviations in the P-Q relationship, e.g. Tijdeman et al., 2018.*

(3) This will be added in.

*(1) P2, Intro. The paper would do well to refer to the expansive literature in hydroecology which also tackles a similar problem of estimating 'natural' flows for sites, against which impacted flows can be compared. The classic papers of Brian Richter are a good start, and I'm fairly sure methods have been proposed to transfer natural flow percentiles (but using a whole FDC approach; try the DHRAM work by Andrew Black, Dundee as a start). Another area where this is done routinely is through the LowFlow software produce, a regionalisation product which estimates natural and disturbed FDCs at any site. Its not drought specific, but definitely has a very similar aim.*

(2) We agree that literature from the hydroecology field can be brought in to show how they address the similar problem. We thank the reviewer for the suggestion and the references, and we will update the introduction.

*(1) P4, Sect. 2.3 Given the concerns raised about the UK catchments, this section needs to be reconsidered.*

(2) This section will be changed in the revised manuscript to introduce the new case study, Blackwater, and update all associated sections, tables (Table 2; Table 3) and figures (Figure 3).

*(1) P6, L2. The 80th percentile is not what is being used here. This paper uses the 20th percentile, or, as is most commonly referred to in hydrology, Q80: the 80% non exceedance threshold from the flow duration curve.*

(2) The authors agree that this needs rewording, and will be for the revised version throughout.

Existing phrasing of: "Here the 80$^{th}$ percentile was used as the threshold, meaning that 80% of the time discharge is above the threshold. The 80$^{th}$ percentile is a commonly used threshold to identify drought events (Hisdal & Tallaksen, 2000; Fleig et al., 2006; Heudorfer & Stahl, 2016)."

Has now been changed to:

"Here the 80% non-exceedance threshold (Q80) from the flow duration curve was used. This means that 80% of the time discharge is above this threshold. The Q80 is a commonly used threshold to identify drought events (Hisdal & Tallaksen, 2000; Tallaksen and Van Lanen, 2004; Fleig et al., 2006; Heudorfer & Stahl, 2016)."

*(1) Discussion: is generally very insightful but definitely needs reconsidering in light of catchment selection issues, and claims about attribution needed to be moderated.*

(2) We thank the review for their comment and the suggestion to highlight the limitations and issues further. We will change the discussion to include more about these aspects.

We thank the reviewers for all their comments and suggestions, and we hope that a number of them have been satisfied by the change of UK catchment, and that we have strengthened the paper with the suggestions enhanced analysis and provided a more clear aim (i.e. a methodological paper rather than a case-study type of paper).

**References**

AEDA, 1990. The Ely Ouse Essex water transfer scheme, Agricultural & Environmental Data Achieve. http://www.environmentdata.org/archive/ealit:4162

BBC, 2011: "Thames Water accused over dry River Kennet in Wiltshire", BBC, 27 September 2011, https://www.bbc.co.uk/news/uk-england-wiltshire-15076406

Kennet catchment partnership: http://www.kennetcatchment.org/issues/over-abstraction/

Rangecroft, S., Van Loon, A. F., Maureira, H., Verbist, K., Hannah, D. M. 2016. Multi-method assessment of reservoir effects on hydrological droughts in an arid region, ESDD, doi: 10.5194

Robinson, W. 2011. Essex & Suffold Water: Ely Ouse to Essex Transfer Scheme. https://www.raeng.org.uk/RAE/media/General/News/Documents/20111027-William-Robinson.pdf

Tallaksen, L.M. and Van Lanen, H.A.J. (Eds.) (2004) Hydrological Drought. Processes and Estimation Methods for Streamflow and Groundwater. Developments in Water Science, 48, Elsevier Science B.V., 579 pg.

Tallaksen, L.M., Hisdal, H. and Van Lanen, H.A.J. (2009) Space-time modeling of catchment scale drought characteristics. J. of Hydrol. 375:363-372 (doi:10.1016/j.jhydrol.2009.06.032).

Tijdeman, E., Hannaford, J. & Stahl, K. 2018. Human influences on streamflow drought characteristics in England and Wales, HESS. Vol 22(2): 1051-1064.

Van Loon & Van Lanen, 2013. Making the distinction between water scarcity and drought using an observation-modeling framework. Water Resources Research, vol 49(3): 1483-1502.

Van Loon & Van Lanen, 2015. Testing the observation-modelling framework to distinguish between hydrological drought and water scarcity in case studies around Europe. European Water, 49: 65-75.

Wanders, N. & Wada, Y. 2015. Human and climate impacts on the 21st century hydrological drought. Journal of Hydrology, vol 526: 208-220.

---

## Author Response (AR1)

**Editor**

Dear authors,

As you have seen, the two reviewers provided detailed comments on your manuscript. They highly appreciate the general research idea and direction. I fully agree with them.

However, they both also highlight the critical need for an in-depth reflection on the choice of catchments for the paired analysis. Considerable doubts are expressed that adequate reference conditions could be established and that the selected catchments do not exhibit sufficient similarity to allow a meaningful analysis. I strongly encourage the authors to address these issues in detail together with the other comments made by the reviewers. I am looking forward to receiving a revised version of your manuscript

Best regards,

Markus Hrachowitz

*>> Dear editor, thanks very much for your positive evaluation of our research idea and manuscript. We have taken on board the comments of both reviewers and S Mylevaganam.*

*In summary we have changed the terminology of the paper (instead of 'natural' we now use the word 'benchmark' catchment) to reflect the fact that we do not require an undisturbed catchment to compare with the human-influenced one in order to quantify the effect of the human activity we are interested in. We also think that this change of terminology from 'natural' catchment to 'benchmark' helps to show the wider applicability of this approach in the Anthropocene. The most significant change to the paper is that we have changed the UK catchment pair, because we felt that it was too contentious to be used as a neutral example case study with the aim to show a methodology. We now show the application of the paired catchment analysis on different human activities with a case study in which human activities alleviate droughts (UK Blackwater) as well as the existing Australian case study, which shows an aggravation of droughts due to the human activity. We believe that this change strengthens the paper and alleviates the reviewers' comments. We have also introduced a flow diagram (Fig.1 to replace the previous Table 1) to illustrate more clearly the selection of paired catchments. Finally, we have improved the general structure and writing of the manuscript following the reviewers' suggestions and our own thorough re-reading of the manuscript and have re-written the abstract so that it matches again the revised manuscript.*

***Please see below for our response to the reviewers' comments which are given in italics and blue. The page and line numbers mentioned refer to the tracked-changed version of the revised manuscript. We also submit a cleaned version for easier reading of the revised manuscript.***

**Reviewer #1**

*>> We thank you for your review and constructive comments. We hope that you find our responses satisfactory to the points that you make.*

1a) If the authors are trying to determine the presence of human influences on hydrological drought, they should know the natural or reference influence to drought, i.e., the base hydroclimatic and basin characteristics. The authors do a rigorous hydroclimatic analysis in the way that their "paired" basins are same in size, close to each other, have same PET values, same P values. They have no similar geological characteristics but the authors are aware of this.

However, even though as the authors suggest, finding completely natural basins is difficult, their "isolation of the human influence" is coarse and undocumented, without properly accounting for differences of land use and water use between basins and without considering all the land use and water use already existing in the "natural" catchments.

Groundwater abstraction is not the only human effect that can influence drought characteristics. Just by looking in google earth at the Dun River upstream of Hungeford, UK, I see a basin that is completely agricultural, with barely any patch of natural vegetation. Since land cover/use is a control of water partitioning and runoff (Sterling et al., 2013), I don't know how "natural" is this basin. Additionally, by a quick search in google earth, I see that the river is also completely regulated by embankments and check dams/levies from beginning to end. Since flow regulation alters the partitioning of water and hence runoff (Jaramillo and Destouni, 2015), your reference conditions in terms of intra-annual variability and quantity of flow are already altered by regulation.

This means that your natural catchments are already affected and not eligible for a paired analysis for the purpose you need them. I would look harder for other basins with no water use/regulation and more pristine land covers to represent the "natural" condition, in the UK or abroad. Articles such as (Dynesius and Nilsson, 1994; Jaramillo and Destouni, 2015; Lehner et al., 2011; Nilsson et al., 2005) could help. The Authors can also choose instead of a "paired" analysis of catchments a paired analysis of two groups of catchments, this would make the analysis more robust.

*>> The reviewer makes an important comment about classifying the natural catchments as fully natural, and it has made us to reconsider the terminology used in our work to help represent this better. Whilst we fully agree with the reviewers comment that finding a pristine catchment is crucial if one aims to quantify the OVERALL effect of all human influences on drought. However, if we want to use the approach to increase our understanding of how different human activities influence drought we need to be able to isolate a specific type of human influence. Therefore, when comparing a human-influenced catchment with a completely natural catchment, the combined effect of a mix of human activities will be found, which makes it harder to attribute the difference to one type of human influence. It would be extremely difficult to find completely natural catchments that are similar enough to the influenced catchments with only one type of human influence as the difference between them. There are hardly any truly pristine catchments in the UK and most other regions across the world, where there is no water use/regulation and pristine land cover. Any pristine catchments are likely to be incomparable with the 'human' catchments because of their different geology and climate and the human-influenced catchment is likely to have a mix of land-use change, abstraction, etc. To address this, we changed the terminology from 'natural' catchment to 'benchmark' catchment throughout the manuscript and explained more clearly the aim of the approach.*

*With regard to the choice of the UK catchments – The Dun is one of the UK benchmark catchments (Harrigan et al., 2018), therefore it is considered as a 'natural' catchment by the Centre for Hydrology and Ecology and we are confident in using it as a comparison pair. However, we agree that the choice of UK catchment is too contentious to be used as an example, and we have changed the UK example case study pair to accommodate all reviewers' comments. As UK case study, we now present the pairing of the human-influenced Blackwater catchment and the Chelmer (benchmark catchment), both in South-East England. The Blackwater catchment has a water transfer scheme delivering excess water to the Essex area, helping to keep it wet during the summer in dry years (Robinson, 2011). Tijdeman et al. (2018) suggest pairing with a similar catchment, Chelmer, to*

*represent the natural situation with very similar catchment characteristics. Chelmer is identified to have minor artificial influences, with the main difference being the water transfer present in Blackwater (see Section 3.1 in the revised manuscript).*

*With regard to the choice of Australian catchments – we find that the land use in both catchments is similar (see new Figure 3 with a map of catchment's land use). Both catchments have a mix of natural savannahs and grassland used for grazing, with some dryland cropping. According to Green et al. (2011) the Cox catchment land use is "a combination of grazing and dryland cropping with agriculture". The benchmark catchment, Cockburn, is described as "the area is mainly used for grazing with some dryland cropping and horticulture" (Green et al., 2011). The only difference between the catchments is the heavy groundwater abstraction in the Cox (Ivkovic et al., 2014), which is the human influence we are aiming to quantify (see Section 3.2 in the revised manuscript).*

*We have added some of these details about the catchments, their land uses and similarities into the revised paper to show this clearly, especially the change of terminology from 'natural' catchment to 'benchmark' catchment.*

1b) Furthermore, the authors say that one of the basins suffers from groundwater abstractions while the other one doesn't, at least, for the UK pair. There is no quantification of this, how much, where, since when ground water abstraction in the human affected basin? Any evidence that in the other basin there is no groundwater abstraction, are you sure of this?

The authors should check this for the Australian case too.

*>> Unfortunately, we do not have long time series data of actual groundwater abstraction. This is precisely the reason why we are suggesting a paired-catchment approach instead of for example detailed physically-based modelling incorporating detailed data on human activities. Data on human influences in a catchment is generally not available and if it is, it often cannot be shared because it is protected by privacy laws. Of course, qualitative information on the human influence is, however, essential for a successful pairing and attribution of the differences. We have added this to the new Figure 1 and expanded Section 3.1 and 3.2 (of the revised manuscript) to include this qualitative information for our case studies. We changed the UK case study, which now focuses on the effects of water transfer instead of groundwater abstraction. For this water transfer, we have qualitative information about the starting year and purpose of the scheme from AEDA (1990). For Australia, the human-influenced catchment, Cox, is subject to heavy groundwater abstractions (Ivkovic et al., 2014). Although we do not have time series of abstraction, we do know that ACTUAL annual aquifer abstraction rates of the Cox are 11245 ML/yr (Barrett, 2012), whereas LICENCED annual aquifer abstraction licenses for Cockburn are 4481 ML/yr (O'Rourke, 2010). Annual abstractions in the Cox catchment are therefore at least 2.5 times as much as in the Cockburn, but because O'Rourke (2010) reports entitlements only and for the bigger catchment in which the Cockburn is located only 20% of the entitlements is used (O'Rourke, 2010), our estimate is that the actual annual groundwater abstraction in the Cockburn is around 900 ML/yr. The Cox groundwater abstraction would then be more than 12 times as much as the Cockburn.*

2) Include some statistics of land use and water use in the Description of the area. Also, improve Figure 2, to show more information on each catchment.

*>> We have now provided more information about land use in both paired catchments in the text (see new Section 3.1 and 3.2) and changed Figure 3 to visualise land use information for all catchments. We have also added land use as a selection criterion to Figure 1 and the text in Section 2.1 to show that it is a consideration for catchment selection.*

3. The authors should put some statistical significance tests to support further their results.

*>> Whilst this would be a useful suggestion, unfortunately there are not enough data points in our analysis for statistical significance tests. We believe that quantifying the percentage change due to the human influence for the overall drought characteristics and showing the basic descriptive statistics of average, maximum and total duration and deficit volumes already provides very relevant information. We considered doing an event-by-event based analysis, but that proved to be impossible, partly because the input data is not completely similar for both catchments leading to small variations between events. Thus, taking a more holistic look at the average, maximum and total drought duration and deficit over the entire time series is the most robust, and scientifically acceptable, information that we can provide. To perform statistical significance tests we would need more (replicate) catchment pairs, which is beyond the scope of this paper but could be an interesting follow-up.*

So I like the idea, but a more thorough selection of catchments including land and water use conditions needs to be included in the study

*>> Thanks for your summary of the review. We hope that we sufficiently addressed this under items 1 and 2. We believe that adding this information will help to provide the right level of detail for the reader, and the change of terminology from 'natural' catchment to 'benchmark' catchment will help to show how we are aiming to isolate human influence to be able to attribute differences to different types of human activities.*

**Authors reply to Reviewer 2's comments**

*Please find our responses and actions to your comments below. Frequently we have placed a number of comments under the same section if they are related and our response covers them jointly. Figures and tables shown here only in this reply have the prefix R (e.g. Figure R1) to distinguish them from those shown in the manuscript. In our response to Reviewer 1, we have suggested that we rename the 'natural' catchment to be termed the 'benchmark' catchment; therefore, the new term is used throughout in our response here.*

*There are a few major things that we changed in the manuscript in general, which we would like to highlight here before discussing the specific comments. We bullet point these major changes here, which all relate to the framing of the paper and the content:*

- *Methodological framework flow diagram. As this is a methods paper, we focus on the methodology aspects and not the results of the case studies. We have introduced a flow diagram to illustrate more clearly the selection of paired catchments.*
- *Demonstrating the method with case studies of different human activities, which alleviate and aggravate droughts. We now show the application of the paired catchment analysis on different human activities with a case study in which human activities alleviate droughts (UK Blackwater) as well as the existing Australian case study, which shows an aggravation of droughts due to the human activity. This changing of the UK paired catchment case study from the Kennet to the new case study of Blackwater addresses a number of Reviewer 2's concerns about the UK pairing.*

**Reviewer's comments**

This is a generally well written paper on an important topic and the paper has significant potential to make a very worthwhile contribution to the 'drought in the Anthropocene' debate. It is potentially a very useful advance arising from a simple yet potentially very effective idea.

*>> We thank you for their overall view of the importance of the topic and paper.*

**1) Major Reviewer's comment: pairing of UK catchment**

Major Comment: impact of catchment (dis)similarity on the proposed method. The method is predicated on the similarity of the donor natural catchment/target influenced catchment, but in the UK example at least, the catchments are not similar enough, very likely leading to over-estimation of the anthropogenic effect.

Catchments are different in terms of runoff response/catchment function, despite their similar rainfall.

Given the more limited range of the Dun flows, my guess is the Q80 of the Dun will be somewhat higher, leading to inflated values for the deviations that are used to infer aggravated drought due to human effects.

*>> We realise that a large proportion of the Reviewer 2's comments are addressing the pairing of the UK case study specifically, highlighting the issue of the (dis)similarity between the UK pair and the contentious nature of this catchment. The reviewer's comments help to show the level of information needed for the proposed paired-catchment method, and the importance of a suitable pairing. We have taken on board some of the reviewer's comments to help improve the pairing.*

*As this is a methods paper and the case studies shown purely demonstrate the method, we found a UK pair that is more similar, as demonstrated by other studies (e.g. Tijdeman et al., 2018), and less contentious, and which helps to demonstrate the use of the paired catchment analysis on a different human activity. Therefore, we have changed our UK case study from the Kennet to the Blackwater catchment (Figure R1). The Blackwater catchment has a water transfer scheme delivering excess water to the Essex area, helping to keep it wet during the summer in dry years (Robinson, 2011). Tijdeman et al. (2018) suggest the pairing with a similar catchment, Chelmer, to represent the natural situation with very similar catchment characteristics. Chelmer is identified to have no artificial water input, with the main difference between the catchments the water transfer present in Blackwater.*

[Figure]

*Figure R1: Discharge plotted (mm/month) for both catchments in new UK case study pair: Benchmark, Chelmer (blue) and human-influenced, Blackwater (red)*

*While the focus of this methods paper is to use the case studies to illustrate the method rather than attributing differences to specific causes, we think that choosing a different and less controversial pairing with more similar runoff response will better highlight the advantages of the method.*

*Please find the rewritten section of the new UK pair in Section 3.1 of the revised paper and updated Table 1, and the new case study results in Table 2 and Figure 3. By changing the UK case study, we hope to have answered the reviewer's questions.*

In terms of the results, the figures quoted in Table 3 seem very large. Given the nature of the debate around abstraction impacts, these figures could be quite contentious, as the Kennet is something of a poster child in the debate around sustainable abstraction, and the authors should do more to ensure they are meaningful.

*>> We agree that the numbers were very large for our Kennet case study, but this might be realistic given the published concern over over-abstraction in the Kennet area, and recent changes to abstraction rates to reduce this (2014 EA issued a notice reducing the permitted abstraction from Axford and revoked the abstraction licence on the little River Og). Therefore it is legitimate that impacts seen over the previous decades could reflect this over-abstraction.*

*e.g. "Thames Water accused over dry River Kennet in Wiltshire", BBC, 27 September 2011 (https://www.bbc.co.uk/news/uk-england-wiltshire-15076406)*

*e.g.2: "For over 20 years there has been local concern that flows in the Upper Kennet above Marlborough have been affected by over-abstraction' (Kennet catchment partnership). (http://www.kennetcatchment.org/issues/over-abstraction/)*

*However, given that the focus of this paper is to introduce the method, we have changed the UK case study to alleviate this comment (see above).*

**2) Overall reviewer's comments: considering other approaches to bolster the method / convincingly demonstrate the method:**

Given these concerns, the authors could consider some approaches to bolster the method and provide verification – e.g., how this method performs relative to other approaches or modifications, e.g. detecting deviations based on rainfall (as in Tijdeman et al. 2018) and PE.

Paired catchment analysis is a staple of experimental hydrology, but is much harder to do in 'real world' examples when it is not possible to control all variables except the main intervention of interest, and even the latter may be poorly understood.

I feel major revisions are needed to convincingly demonstrate the method, through modifiying it to allow some tolerance in the donor/target relationship, verifying the methods using independent abstraction data, or benchmarking it against other methods.

*>> We argue that the paired catchment approach can be extremely beneficial if you have the right level of information to justify the pairing, but not enough detailed data on the human influences themselves (i.e. time series of abstraction rates) or to run alternative approaches. Therefore, the paired catchment approach can help provide insight into the human influence in the catchment from the observation data available and might be a first step to obtain more data to apply alternative*

*approaches. Exploring the use of other methods to obtain more robust results for the example case studies is not the purpose of the paper. In the revised manuscript, we explain that this method can be complementary to others (see p.2 l.20, p.4 l.14-15 and p.15 l.6-7).*

*Furthermore, we have also changed the framing of the paper in terms of the case studies presented. We now show a case study which has a human activity aggravating hydrological droughts (e.g. Australia Cox – groundwater abstractions) but also a case study in which the human activity alleviates hydrological droughts (e.g. UK Blackwater – water transfer). We believe that this helps to demonstrate the method and its applicability better.*

*To help illustrate the method further, we have also added a flow diagram for the pairing of catchments for the analysis, Figure 1.*

In general the approach could be strengthened considerably by taking a more water balance approach as done in the classic paired experimental catchment studies, and also in the study of Prosodcimi which incorporates climate variables to account for any confounding effects.

*>> In the new framing we have emphasised the use of the paired catchment approach when less detailed information is available to still make an assessment of the difference between the two catchments (p.2 l.1-2, p.3 l.18-19). The use of the same time period as a comparison between the two catchments means that overall climate is accounted for in the analysis, which is an advantage compared to methods comparing data pre- and post-disturbance.*

*The water balance approach is useful, but does not focus on drought specifically and requires data that is not available; instead we are looking for a method which can be applied to multiple case studies to give an idea about the impact of human activity on drought only based on commonly available data (precipitation and discharge).*

**3) Reviewers comment: include abstraction rates information**

verifying the method using independent abstraction data, or benchmarking it against other methods.

The Kennett is very well known to experience major abstractions, which have been non-stationary over the series. But more could be done to follow this study up – there is anecdotal information on abstractions in various grey literature sources I found online (below).

*>> As mentioned, some average annual abstraction values are available for the case studies through grey literature. However, it is difficult to obtain actual abstraction, rather than licenced values and long time series of actual abstraction are impossible to obtain. The paired catchment method is therefore valuable when other methods cannot be applied because quantitative abstraction information is not available. We do agree that qualitative information is needed for the selection of a suitable pair and to interpret results (see new Figure 1). We have therefore added more information to the manuscript (see Section 3.1 and 3.2), also in response to the comments of Reviewer 1 (1-2).*

**4) Reviewers comment: Impacts and abstraction data**

Finally, to really demonstrate the success of the method, it would be nice to have some independent verification of the suggested impacts. I appreciate access to abstraction data is not straightforward for the UK, but might be possible for one catchment, at least for derived data on impacts rather than particular abstractions.

*>> We would like to be careful not to allocate too much of the paper to the two case studies as we feel that this will detract away from the focus of the paper, which is the method itself. However, we have included more information here to give a background on the new UK pairing (see new Section 3.1). It is also important to note that gaining the level of information we have for the UK is possible, but it is difficult to get similar data for many other regions of the world, including Australia.*

I would suggest some dialogue with the EA would be worthwhile, as there seem to be naturalised data (be decomposition and/or modelling) available for the Kennet for various part studies.

*>> We fully agree that a comparison with other methods that rely on modelling is interesting. However, we do not feel that the EA naturalised data can be used as a benchmark or independent verification to compare the paired-catchment results against. Modelling has large uncertainties and especially for the EA naturalisation it is not clear which processes and activities are included in the analysis. Additionally, naturalised data is not available for the Australian catchments.*

**5) Reviewers comment: using threshold from benchmark catchment**

However, transferring a threshold directly from one catchment (reading off the Q80 flow value from the natural catchment and applying it to the influenced one) to another seems like a potentially dangerous business. This might not be a problem if one is just trying to estimate flows at an ungauged site, and can report uncertainties; but in the present method any biases arising from the data transfer could be very misleading.

*>> The transferring of thresholds from the donor catchment to the influenced catchment is a basic principle of this application of the paired-catchment approach. Furthermore, the transferring of thresholds from a benchmark situation to a human-influenced situation has been used in other existing literature, with regards to the comparison of naturalised data and observed flow (observation-modelling framework, Van Loon & Van Lanen, 2013; Van Loon & Van Lanen, 2015). It is also used by the large-scale hydrological modelling community when analysing future droughts – use of pristine threshold for both the pristine and human scenarios to calculate the modelled human impact on hydrological droughts (e.g. Wanders & Wada, 2015).*

*The main issue with using a threshold established on the human-influenced catchment discharge is that the effect of the human activities is then included in the threshold used to calculate droughts. We feel that this is even more misleading as it underestimates the effect of the human influence. For a fair quantification of the effect of human influence on drought we therefore adhere to using the same point of reference (the same threshold) for both catchments.*

**6) Reviewers comment: Impact of catchment (dis)similarity on the proposed method**

Put simply, the method applied in this paper can only work if the donor's natural flow regimes is near-identical to the 'theoretical natural' flow regime of the target site. Any deviation between these regimes will be interpreted as anthropogenic; when it could just be due to the variations between two catchments that appear quite similar but are in fact different.

It is difficult to find suitable pairs. Even when catchments are in principle very similar (geology, rainfall etc), the concept of 'uniqueness of place' is a major obstacle.

*>> We agree that 100% proof of similarity cannot be given. Experimental hydrology (control and treated catchments/plots) and models are also unable to provide this. We can only reliably check the similarity of both catchments if we have long time series (preferably 30 years) for the donor and the human-influenced catchment prior to human disturbance. In practice, this will be very rare or non-*

*existing for catchment pairs. We now mention this aspect more clearly in the revised manuscript (e.g. p.1 l.17, p.6 l.18-23, p.14 l.18-26).*

As a result, I do not think the authors can claim 'attribution', and the claims of the paper need to be reconsidered.

Note that this an important difference in the urbanisation paper (Prosdocimi et al., 2015) or in the classic experimental catchments, which all incorporate some data on the intervention in question into the analysis (e.g. the land cover data used by Prosdocimi et al.).

*>> We agree and we have rephrased our results from attribution to explanation or isolation throughout the manuscript. We have also covered this topic in our revised Discussion section (p.15 l.3-7) and Conclusion (p.15 l.32).*

**Specific reviewer's comments**

A technical matter: In Figure 3, I'm surprised to see so little of the flows being below the threshold. It does not look like 20% of the flows are below the threshold to me – can the authors please check?

*>> We can confirm that it is correct. The total number of months in drought (Table 3) is close to 20%: 93 months in drought out of 540 months (17%). It is not the full 20% because we have dropped minor droughts that were only 1 month in duration.*

P2, L21. Another approach is using deviations in the P-Q relationship, e.g. Tijdeman et al., 2018.

*>> This has been added in (p.3 l.33 – p.4 l.2).*

P2, Intro. The paper would do well to refer to the expansive literature in hydroecology which also tackles a similar problem of estimating 'natural' flows for sites, against which impacted flows can be compared. The classic papers of Brian Richter are a good start, and I'm fairly sure methods have been proposed to transfer natural flow percentiles (but using a whole FDC approach; try the DHRAM work by Andrew Black, Dundee as a start). Another area where this is done routinely is through the LowFlow software produce, a regionalisation product which estimates natural and disturbed FDCs at any site. Its not drought specific, but definitely has a very similar aim.

*>> We agree that literature from the hydroecology field should be brought in to show how they address the similar problem. We thank the reviewer for the suggestion and we have updated the Introduction (p.3 l.6-8, p.6 l.20-23) and the Discussion (p.14 l.23-26).*

P4, Sect. 2.3 Given the concerns raised about the UK catchments, this section needs to be reconsidered.

*>> This section has been changed in the revised manuscript to introduce the new case study, Blackwater, and update all associated sections (Section 3.1), tables (Table 1, 2) and figures (Figure 3, 4).*

P6, L2. The 80th percentile is not what is being used here. This paper uses the 20th percentile, or, as is most commonly referred to in hydrology, Q80: the 80% non exceedance threshold from the flow duration curve.

*>> The authors agree that this needed rewording, and this has been done in the revised version of the manuscript (p.9 l.17-20).*

Discussion: is generally very insightful but definitely needs reconsidering in light of catchment selection issues, and claims about attribution needed to be moderated.

*>> We thank the reviewer for their comment and the suggestion to highlight the limitations and issues further. We have changed the discussion to include more about these aspects (see Section 4).*

*We thank the reviewer for all their comments and suggestions, and we hope that a number of them have been satisfied by the change of UK catchment, and that we have strengthened the paper with the suggestions enhanced analysis and provided a clearer aim (i.e. a methodological paper rather than a case-study type of paper).*

**Short comment by S Mylevaganam**

*We thank you for your review and constructive comments. We hope that you find our responses satisfactory to the points that you make.*

1) As per the authors, research using the paired catchment approach to assess change in hydrological "droughts" due to land use and other human activities remains limited(see P-3 LN-29). However, as per the authors, research works using the paired catchment approach to assess "low flows" due to land use and other human activities are found in the literature(see P-3 LN-24:29). What are low flows? What are droughts? I think, the distinction between low flows and droughts needs to be explicitly mentioned in the manuscript to assert the authors' statement that the research using the paired catchment approach to assess change in hydrological "droughts" due to land use and other human activities remains limited. Moreover, what is implied by "limited"? Should the authors include/cite the research works using the paired catchment approach to assess change in hydrological "droughts" due to land use and other human activities found in the literature?

*>> We thank the reviewer for pointing out this for clarification. In the original manuscript, we defined droughts at the end of the first paragraph (revised manuscript p.2 l.31 – p.3 l.2). However we have now included a definition of low flows (p.5 l.21-22) and added a brief description of how these two terms differ to make it clearer to the reader (p.5 l.28-29).*

*By "limited" we mean that there are actually no published articles specifically assessing effects of human activities on hydrological droughts using paired catchments. Tijdeman et al. (2018) use the concept of comparing droughts occurring in two similar catchments (e.g. Supplementary figure S4). They do not use a specific framework as we present here, but they explicitly state the potential use for it: "An alternative approach is based on the principles of paired catchment analysis, a concept that has been a foundation of process hydrology. Typically, a paired catchment study compares the flow regimes of nearby catchments with similar physical characteristics. The approach has been applied in numerous iconic experimental studies to investigate land use impacts on river flow (e.g. review of Brown et al., 2005). However, the paired catchment concept can also be used to study human influences on streamflow, using existing gauging station networks, if appropriate "donor" natural catchments with similar flow regimes can be found for "target" catchments with known influences (as conducted in the case of urbanisation effects on floods; Prosdocimi et al., 2015)." (Tijdeman et al., 2018, p.1053). We have changed the statement in the manuscript to demonstrate that there are no published articles specifically assessing human activities on hydrological droughts using paired catchments (p.5 l.29-30).*

2) As per the authors, drought analysis is normally conducted on the daily or monthly time step (see P-5 LN-23). Therefore, the authors use "monthly" data for the paired catchment analysis, even

though the selected catchments are provided with data on daily time step? In the current version of the manuscript, the authors fail to state the reason for using "monthly" data for the paired catchment analysis. Is it the methodology (i.e., paired catchment) that is chosen in the proposed approach forces the authors to use the monthly data instead of daily data? Should the authors show the characteristics of the droughts within a month for the selected catchments? What is the minimum duration (in days) of the drought observed in the selected catchments? What is the minimum frequency (in days) of the drought observed in the selected catchments? Would not these details justify the applicability of selecting the monthly time step to conduct the drought analysis?

*>> As stated in the manuscript (p.8 l.30 – p.9 l.2), drought analysis is commonly done on the monthly time step, therefore it is deemed justifiable. Droughts are generally long phenomena with timescales of weeks to years, with drought-generating processes and associated impacts going beyond a daily time step.*

*Furthermore, monthly data was used to avoid a need for pooling of drought events, and to allow minor droughts of <1 month to be dropped. Ultimately as long as the same time resolution and threshold is used for both catchments then the comparison remains valid.*

3) The use of paired catchment in the proposed approach is very much subjective. Is it possible to define hydrologically similar catchments without considering the landuse pattern and spatial orientation of the landuse? Would it be possible to completely define hydrologically similar catchments using precipitation, PET, and geology? The landuse pattern than the spatial orientation of the landuse may dramatically alter the flow pattern even for the same precipitation, PET, and geology?

*>> We do consider land use (see reply to comment 2 of Reviewer 1) and land use is similar for the catchment pairs. We have now included this catchment characteristic into Figure 1 and 3 to show the importance.*

4) In Table 3, for Dun (natural catchment), how did the authors compute the total number of months in drought and the frequency? Since the authors have analyzed from 1973 to 2013, there are 12*41(=492) monthly flow records. In other words, setting 80% as the threshold level yields around 98 months in drought (see Table 3). What is the definition of frequency? Is it meant for the return period of the drought? What is the unit of frequency?

*>> Because minor drought events of 1 month have been dropped, it might result in slightly less drought events being identified than expected. However, our previous results show that the Dun has 93 months in drought, which is extremely close to this expected 98 months, and the differences seen between the two numbers is purely because of the dropping of minor drought events.*

*Frequency = number of drought events identified in the time period. This definition is now included in the revised version of the manuscript (p.9 l.31-32). There is no unit for frequency as it is just a value representing the number of drought events identified.*

5) In the current version of the manuscript, the authors evaluate the groundwater abstraction on the droughts. With groundwater abstraction, it is expected to have a depleted groundwater table. Consequently, as per Darcy's law, since one of the drivers (i.e., hydraulic gradient) is changed due to groundwater abstraction, a possible alteration on the base flow (see the reported BFI values in UK) that defines the low flow conditions in most of the rivers is expected. Would it be possible for the authors to show the rainfall pattern (daily) in the natural catchment and human altered catchment for some of the drought periods in UK?

*>> Indeed groundwater abstraction leads to lower groundwater heads, decreased groundwater gradients, lower groundwater inflow/base flow into the rivers, and hence more extreme drought in streamflow. It is important to note that we do not take an event-by-event analysis though (as it is a challenge to pair drought events in different catchments), but look at overall averages, maximums and totals of drought duration and deficit over the entire time series to avoid this level of discretion.*

6) As per the authors, the paired catchment approach has been a predominant method for detecting the effects of disturbance on catchment scale hydrology (see P-3 LN-14). To support this statement, the authors cite Zégre et al., 2010. Considering the fact that the paired catchment approach has been used for many decades (see P-3 LN-16), starting as early as 1920 (see P-3 LN-19), would it possible for the authors to state the reason for citing Zégre et al., 2010 to support the statement(see P-3 LN-14)

*>> Thanks for this comment. In our revised version we have changed this citation to one of an earlier time period, that of Bates (1921), which is the first paired-catchment study (p.4 l.29-30).*

7) Should the section 2.1 be re-written? The third paragraph of section 2.1 is about the method (i.e., paired catchment approach). In other words, the third paragraph of section 2.1 defines the method. However, the first and the second paragraphs of section 2.1 briefly summarize the crux of some of the previous research works using the paired catchment approach found in the literature. Does it make sense to mention the previous research works using the paired catchment approach at first and then define the method? From the reader's point of view, the third paragraph of section 2.1 should come first and then the previous research works using the paired catchment approach found in the literature.

*>> This restructuring of section 2.1 is done in the revised version (see new Section 1.2).*

8) The section that is devoted for discussion (i.e., section 4.0) is structured using some of the limitations of the approach presented in the manuscript. What is expected in the section (i.e., discussion) by a reader of the manuscript is left missing in the current version of the manuscript.

*>> We believe that because the aim of the paper is to present the novel application of a simple framework for quantifying the human influence on hydrological droughts, this is the important aspect to be discussed in this section, rather than the results of the two case studies which are used to demonstrate the method. Discussing the limitations and the possible ways forward for future research with this method is useful for future research, given its scalability to be performed in other locations.*

9) From the reader' point of view, the numbering of the sections is misleading. In section 2.2, the authors introduce the approach used in the current version of the manuscript to address the intended tasks (i.e., determination of drought metrics). However, in section 2.2, the authors ends the paragraph (see P-4 LN-17: here we outline the important elements for the "approach") to form the subsections that need to be listed under section 2.2. Should the sections 2.3, 2.4, and 2.5 be numbered as section 2.2.1, 2.2.2, and 2.2.3?

*>> We agree that the numbering here could be improved. We have made Section 1.2 "Paired-catchment analysis" its own section within the introduction, followed by Section 1.3 "Aim". This enables Section 2 onwards to focus on the specific application of the method and the relevant information for the analysis.*

10) On P-3(see LN-22), what is meant by "see review of Brown et al., 2005"? Is it a review about Brown et al., 2015? On P-3(see LN-24), what is meant by "some studies included low flows"?

*>> We are referring to the published review of paired catchments written by Brown et al. (2005) titled 'A review of paired catchment studies for determining changes in water yield resulting from alterations in vegetation', however we have changed the phrasing of this to be clearer (p.5 l.17-19).*

*Most paired catchment studies look at annual changes following treatment, and often focus on water yield and high flows, not on low flows or streamflow during droughts. Therefore we specifically point out that some look at low flows, but not the majority of the published literature. P.5 l.20-30 is an entire paragraph which summarises and points to the work which does use paired catchments to look at low flows.*

**Minor Comments:**

a) The authors' marriage to some of the words (e.g., "here" we suggest, "here" we outline, we "here" give, "here" we use, "here" the 80% percentile, "here" we have analyzed, "here" we present, "here" we have demonstrated, "here" we focused, "here" the focus was, and "here" we show) is a little off from what is expected in a scientific research paper.

*>> We have endeavoured to address this repetition throughout the revised version of the manuscript.*

b) In Table 1, the title of the second column (i.e., "assessment for similarity") needs to be changed to reflect the cell values.

*>> Table 1 has been replaced with a new Figure 1.*

c) In Table 2, the widths of the columns (e.g., column-7) need to be adjusted to fit the content.

*>> This table has been adjusted to fit the content.*

d) In Table 2, the gage numbers for the selected catchments in Australia are missing (see Figure 2 and Table 2).

*>> Thank you for pointing this out, these have been added to the revised version.*

e) In Figure 3, the label for the x-axis should be "year"?

*>> X axis label have been changed to Year rather than Dates in this revised version.*

f) In Figure 4, the label for the x-axis should be "year"?

*>> X axis label has been changed to Year rather than Dates in this revised version.*

[revised manuscript text omitted]

---

## Author Response (AR2)

**Editor:**

Dear authors,

thanks a lot for the detailed revisions of the original manuscript, which considerably strengthened the paper. In spite of appreciating your efforts, the two reviewers are not yet fully convinced of your approach. They both highlight a few further points that require some more attention. This relates on the one hand to the issue of regulation: please provide some indications on the level of regulation in the groups of basins. On the other hand, I strongly encourage you to provide a somewhat stronger discussion of the fundamental limitations of paired catchment approaches and the potential consequences of that. You mention on p.3,l.3 that paired catchment approach was "successfully" applied in the past. "Successful" is somewhat exaggerated, in particular in the light of the issues discussed by the critique provided by Alila et al. (2009).

I am very much looking forward to receiving a revised version of your manuscript! Best regards, Markus Hrachowitz

References:

Alila, Y., Kuraś, P. K., Schnorbus, M., & Hudson, R. (2009). Forests and floods: A new paradigm sheds light on age-old controversies. Water Resources Research, 45(8).

*>> Dear editor,*

*Thanks for your suggestions for improving our manuscript. We have now addressed the issues that were raised. We included information on dams and other water infrastructure for both case studies (p.9 l.17-19 and p.10 l.10-11 in the tracked-changes document) and added sentences on the importance of similarity in human activities besides the human activity under study (e.g. p.1 l.26-27, p.6 l.31-32, p.7 l.11-13, p.11 l.26-27). To address the issue of criticism on the paired-catchment approach, we added additional discussion on the paired-catchment approach in the Introduction and Discussion sections (p.4 l.14-19, p.12 l.19-29) and removed the qualification that the paired-catchment approach has been applied "successfully" in the past (p.3 l.22, p.4 l.11). The Alila et al. (2009) paper actually supports our application of the paired-catchment approach, i.e. using a frequency-based analysis, so we also included some statements supporting our choice of analysis (e.g. p.5 l.32 – p.6 l.2, p.8 l.1-2, p.8 l.12-13, p.12 l.19-22). Finally, we made a case for clearly explaining the methodology when interpreting results of paired-catchment studies (p.12 l.28-29). We hope that these changes to the manuscript satisfy your requests and look forward to hearing your evaluation.*

*Best regards,*

*Anne Van Loon and co-authors*

**Reviewer #1:**

I thank the authors for addressing most of my comments and I see that the manuscript has been greatly improved by changing some basins and giving more information on the basins. The methodology is now simple to replicate and straightforward.

*>> Dear Reviewer 1,*

*Thanks for your positive evaluation of our revised manuscript. We have now addressed your remaining issues.*

Two remaining comments:

1. I agree that finding completely natural catchments without human influence and hydroclimatic data availability is difficult. That is why, using a large number of catchments instead of a single pair-to-pair analysis can also yield interesting general patterns pointing to a specific human driver or several (something like "paired groups of catchments" approach). I think this is the approach Nr. 4 mentioned by the authors. However, I think that studies using this approach at a larger scale and that have also studied drought/aridity from the perspective of evapotranspiration (via de Budyko framework) could also be included to complete the literature review.

See for example:

*Destouni, G., Jaramillo, F., & Prieto, C. (2013). Hydroclimatic shifts driven by human water use for food and energy production. Nature Climate Change, 3(3), 213–217. https://doi.org/10.1038/nclimate1719

or

*Jaramillo, F., & Destouni, G. (2015). Local flow regulation and irrigation raise global human water consumption and footprint. Science, 350(6265), 1248–1251. https://doi.org/10.1126/science.aad1010

*>> We agree with the reviewer on the usefulness of studying large numbers of catchments, as suggested in our approach no.4 on page 3. We extended our description of approach no.4 (p.3 l.10-12 in the tracked-changes document), included the references suggested by the reviewer (e.g. p.2 l.27-28, p.10 l.24), and mention the large-sample approach as one of the methods that the single catchment-to-catchment approach, as adopted in our study, could complement (e.g. p.2 l.6, p.12 l.18).*

2. "water use" is an important aspect that definitely cannot be ignored when using a pair catchment approach (Page 10 line 9). And again, I would definitely recommend a comparison on the amount of regulation in both basin or groups of basins used, in terms of for instance, number of embankments, check dams, dams, etc), or any data available on the subject, just to be sure that the benchmark catchment is not heavily regulated. If no data is available, this can still be even done from Google Earth or so. I say this precisely because of the findings of the studies mentioned in my point 1.

*>> We included information on dams and other water infrastructure for both case studies (p.9 l.17-19 and p.10 l.10-11 in tracked-changes document) and added sentences on the importance of similarity in human activities besides the human activity under study (e.g. p.1 l.26-27, p.6 l.31-32, p.7 l.11-13, p.11 l.26-27). There are no check-dams in these case studies and we do not feel that embankments, levees and other flood defences have an important influence on low flows and drought and have therefore chosen not to mention these in our manuscript.*

**Reviewer #2:**

The idea behind this paper is sound, however I feel as though this paper is presenting an argument that will bring us full circle on the debate of what methods are appropriate for understanding the incremental impact that humans have on the hydrological extremes. We have many of the modelling

methods (that are time consuming and full of uncertainty) because of the issues with the paired catchment approach, and their associated uncertainties. This is especially true when we try to interpret our impact on the extremes. See the debates that arose from the issues that Alila and others brought forward around the possible forestry impacts on flooding. The previous reviews highlighted several issues with the original submission, but perhaps focused on the issue of matching the catchments. That is only one source of uncertainty when using the paired-catchment approach. As the authors point out this method has been around for about a century, and unfortunately, the debates about how to interpret data and how transferable the results are is still ongoing. These debates are over data measured in the most controlled environments (very small neighbouring watersheds) with a full BACI design.

I agree with the authors in that we should not let go of the fundamentals of using observations, but the paper does not convenience me that these methods are going to provide new insights. I'm not sure the way forward on this issue, however as it relates to the current paper I might suggest that there is nothing fundamentally wrong with the methodology, it just has lots of uncertainty - which the authors point out. This makes it very hard to have a paper with the objective to propose and justify the use of this method. Making the focus of the paper on results might be an option. Rather than using case studies to highlight this as an acceptable method, perhaps the paired catchment methods can be applied to a region that has existing studies on droughts to provide an additional body of evidence towards (or against) what other studies have presented on the possible human impacts on drought?

*>> Dear Reviewer 2,*

*Thanks for highlighting the soundness of our idea and methodology and for discussing the limitations of the paired-catchment approach. We totally agree that we should recognise the debate on the use of the paired-catchment approach and interpretation of its results, but we disagree that because of this debate the paired-catchment approach cannot be used at all. Given the limitations and uncertainties in all approaches, we should use every approach available, including paired catchments, which has not been applied to distinguish the human influence on hydrological drought before. In our manuscript we present the paired-catchment approach as an alternative approach to complement other methods (p.2 l.5-7, p.3 l.25-26, p.12 l.17-18 of the tracked-changes document).*

*There are many studies that show the benefits of paired-catchment analysis when a full Before-After-Control Impact (BACI) set up is not possible. For example, Andréassian et al. (2012) found that the paired-catchment approach performed just as good or even slightly better than rainfall-runoff modelling for predicting discharge in France (p.4 l.15-16) and Jaramillo & Destouni (2015) argue that modelling is known to underestimate the human influence on hydrology. We think that our use of the Control-Impact (CI) set up is acceptable, because Peñas et al. (2016) found that the CI set up gave satisfactory results for over 80% of the impacted hydrological variables, compared to the BACI set up (p.12 l.2-5).*

*The paper by Alila et al. (2009) is interesting, but it actually supports our approach rather than invalidating it. The authors critique the methodology used in paired-catchment studies of the deforestation effects on flooding, namely the one-to-one pairing of events based on driving meteorology (chronological pairing). They show that this is very dependent on meteorology and antecedent conditions to be exactly similar between catchments and argue that you should instead look at all flood events and consider changes in magnitude AND frequency (frequency-pairing). This is exactly what we do (looking at changes in frequency, deficit and duration over the*

*whole time period, not matching specific drought events), so in the revised manuscript we used this paper by Alila et al. (2009) to support our choice (e.g. p.5 l.32 – p.6 l.2, p.8 l.1-2, p.8 l.12-13, p.12 l.19-22).*

*In the revised manuscript, we also added additional discussion on the paired-catchment approach in the Introduction and Discussion sections (p.4 l.14-19, p.12 l.19-29). In these paragraphs we clarified that there is a debate on the application of the paired-catchment method and the interpretation of the results, and that we recognise that for droughts as well there are different methodological decisions to be made that can influence the results. We now also refer to the comment by Birkinshaw (2014) that the most important is being open about the method you use and recognising its strengths and limitations (p.12 l.28-29).*

[revised manuscript text omitted]